# The complex Liouville string: Worldsheet boundaries and non-perturbative effects

Scott Collier[1]⋆, Lorenz Eberhardt[2]†, Beatrix Mühlmann[3,4]‡ and Victor A. Rodriguez[5,6]∘

**1** Center for Theoretical Physics, Massachusetts Institute of Technology,
Cambridge, MA 02139, USA
**2** Institute for Theoretical Physics, University of Amsterdam,
PO Box 94485, 1090 GL Amsterdam, The Netherlands
**3** Department of Physics, McGill University Montréal, H3A 2T8, QC Canada
**4** School of Natural Sciences, Institute for Advanced Study, Princeton, NJ 08540, USA
**5** Joseph Henry Laboratories, Princeton University, Princeton, NJ 08544, USA
**6** Department of Physics, University of California, Santa Barbara, CA 93106, USA

⋆ sac@mit.edu , † l.eberhardt@uva.nl , ‡ beatrix@ias.edu , ∘ varodriguez@ucsb.edu

## Abstract

We investigate general observables of the complex Liouville string with worldsheet boundaries. We develop a universal formalism that reduces such observables to ordinary closed string amplitudes without boundaries, applicable to any worldsheet string theory, but particularly simple in the context of 2d or minimal string theories. We apply this formalism to the duality of the complex Liouville string with the matrix integral proposed in [1,2] and showcase the formalism by finding appropriate boundary conditions for various matrix model quantities of interest, such as the resolvent or the partition function. We also apply this formalism towards the computation of non-perturbative effects on the worldsheet mediated by ZZ-instantons. These are known to be plagued by extra subtleties which need input from string field theory to resolve. These computations probe and uncover the duality between the complex Liouville string and the matrix model at the non-perturbative level.



# 1 Introduction

In our previous papers [1,2] we introduced the complex Liouville string, which is the critical string theory defined by the following worldsheet conformal field theory

$$
\begin{array}{ccccc}
\text{Liouville CFT} & & (\text{Liouville CFT})^* & & bc\text{-ghosts} \\
c^+ = 13 + i\lambda & \oplus & c^- = 13 - i\lambda & \oplus & c = -26
\end{array} . \tag{1.1}
$$

Here $\lambda \in \mathbb{R}_+$ is a real-valued parameter. In this definition the $c^+$-Liouville CFT denotes the two-dimensional Liouville conformal field theory whose CFT data is defined by analytic continuation of that of $c \geqslant 25$ Liouville CFT to complex central charge.

**Perturbative closed string amplitudes.** The most familiar observables in this critical worldsheet theory are the closed string amplitudes, defined as integrated worldsheet CFT correlation functions over the moduli space of Riemann surfaces $\Sigma_{g,n}$ with genus $g$ and $n$ punctures. In [1], the non-perturbative solution of Liouville theory as well as symmetry properties of the combined worldsheet CFT (1.1) were leveraged to directly calculate specific instances of perturbative closed string amplitudes at low genus. This analysis led to the uncovering of a rich structure in terms of a dual two-matrix model in [2], which in particular facilitated a recursive and explicit representation of the amplitudes $\mathsf{A}_{g,n}^{(b)}(p_1,\ldots,p_n)$ for arbitrary genera and punctures, hence solving the theory at the level of closed-string perturbation theory. However this correspondence has so far only been established perturbatively. In this work we will investigate non-perturbative aspects of this duality. This paper is an expanded version of the corresponding section of [3].

**Other observables.** As anticipated in [1,2] and further developed in [4,5], the interpretation of the critical worldsheet string (1.1) as a theory of 2d quantum gravity, together with the duality with the two-matrix model of [2], motivates the consideration of additional types of observables. In this context, for instance, in related theories such as JT gravity [6,7], the minimal string [8–12], and the Virasoro minimal string [13], a natural observable often discussed is the partition function associated with a spacetime with $n$ asymptotic boundaries and genus

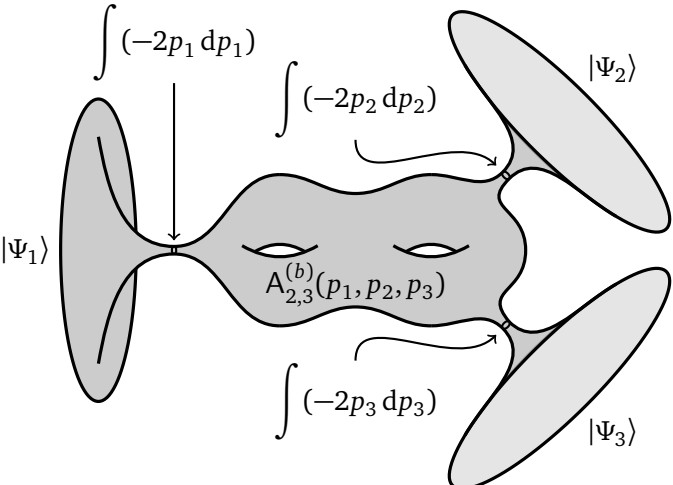

Figure 1: General multi-boundary observables are computed by gluing trumpet partition functions $Z^{(b)}_{\text{trumpet}}(\Psi_j, p_j)$ associated with the worldsheet conformal boundary conditions onto the corresponding closed string amplitudes $A^{(b)}_{g,n}(\boldsymbol{p})$. The conformal boundary conditions denoted by the boundary states $|\Psi_j\rangle$ may correspond for example to resolvents or partition functions of the matrix model, or ZZ-instantons.

$g$ in the bulk. In the recent literature these have been given a holographic interpretation in terms of the perturbative contributions to products of thermal partition functions of a dual quantum system, with inverse temperatures set by the renormalized length of the asymptotic boundaries. We will see that these partition functions are straightforwardly related to the closed string amplitudes $A^{(b)}_{g,n}$ through a specific integral transform.

In order to describe this and other kinds of observables in the complex Liouville string, we will need to consider the inclusion of conformal boundary conditions in the worldsheet CFT (1.1). Indeed, in the examples mentioned previously, the FZZT boundary condition in Liouville CFT was found to play a significant role in the discussion of partition functions [11, 13, 14]. In this paper, however, we will consider the addition of conformal boundaries in a much more general sense — applicable in principle to any worldsheet string theory, but especially simple in this class of "minimal" low-dimensional string theories such as the complex Liouville string (1.1).

In this paper, we will consider a more general class of observables that we will denote by $Z^{(b)}_{g,n}(\Psi_1, \dots, \Psi_n)$ and refer to as a partition function on a Riemann surface with $g$ handles and $n$ boundaries, each associated to (possibly distinct) conformal boundary conditions of type $\Psi_j$. See figure 1 for a schematic picture for the case $g = 2$ and $n = 3$. A specific type of "trivial boundary condition", explained further in section 2.1, reduces the partition function $Z^{(b)}_{g,n}(\boldsymbol{\Psi})$ back to the usual string amplitude $A^{(b)}_{g,n}(\boldsymbol{p})$. By allowing the boundary conditions $\Psi_j$ to be different and non-trivial for each boundary, we obtain what at first glance appears to be a much wider class of observables of the theory. In fact, we will see that the partition functions $Z^{(b)}_{g,n}(\boldsymbol{\Psi})$ with $\boldsymbol{\Psi} = (\Psi_1, \dots, \Psi_n)$ can be constructed from the string amplitudes by gluing suitable "trumpet" partition functions[1] in the following form,

$$Z^{(b)}_{g,n}(\Psi_1, \dots, \Psi_n) = \int \left( \prod_{j=1}^{n} (-2p_j \, dp_j) \, Z^{(b)}_{\text{trumpet}}(\Psi_j, p_j) \right) A^{(b)}_{g,n}(p_1, \dots, p_n). \qquad (1.2)$$

---

[1]Here we borrow the terminology used in the literature on JT gravity, where a "trumpet" with one end of fixed length characterized by $p_j$ and with another asymptotic boundary of type $\Psi_j$ (typically characterized by the renormalized length $\beta_j$ of the asymptotic boundary) on the other end is glued to a Riemann surface $\Sigma_{g,n}$.

Here, the trumpet partition function $Z_{\text{trumpet}}^{(b)}(\Psi_j, p_j)$ is the disk one-point amplitude of a vertex operator characterized by the Liouville momentum $p_j$ in the presence of the specific conformal boundary condition $\Psi_j$. The integral runs from 0 to $\infty$ along a contour that will be discussed in more detail in section 2. The simplicity of this formula is enabled by the fact that there is no nontrivial mapping class group that mixes the moduli space of the core region with that of the trumpet. Hence all the nontrivial gravitational content of the observable $Z_{g,n}^{(b)}$ is equivalently captured by the closed-string amplitude $A_{g,n}^{(b)}$, which are completely solved by topological recursion of the dual matrix integral. In section 2.2 we will explore several examples of trumpet partition functions of different physical significance, including those associated to resolvents and thermal partition functions of the dual matrix model, and to ZZ-instantons.

Nonetheless, the partition function $Z_{g,n}^{(b)}(\boldsymbol{\Psi})$ constructed as in (1.2) is a simple integral transform of the string amplitude $A_{g,n}^{(b)}(\boldsymbol{p})$. We may hence interpret the observables $Z_{g,n}^{(b)}$ as string amplitudes involving particular linear combinations of the usual closed-string vertex operators. In this sense they do not contain any new information about the theory beyond the elementary closed string amplitudes, although they reveal different aspects of the physics of the theory.

**ZZ-instantons.** A special class of conformal boundary conditions correspond to ZZ-instantons. These are constructed from the ZZ conformal boundary condition [15] in each of the constituent Liouville CFTs of the complex Liouville string (1.1). This class of conformal boundaries has a finite action and a very different physical interpretation, as that of additional saddles of the open-plus-closed string field theory on the ZZ-instanton. They lead to non-perturbative contributions to string amplitudes, which have previously been extensively studied in the context of the $c = 1$ string [16–19], the minimal string [20–22], and the Virasoro minimal string [13]. In particular, the calculation of their effects, or of observables such as the partition function with identical ZZ-instanton conformal boundaries in the language of (1.2), suffers from subtle divergences in the zero-mode sector of the ZZ-instanton open string spectrum. In this paper, we employ the recently developed string field theoretic analysis of D-instantons to systematically compute the leading-order non-perturbative effects of ZZ-instantons in the complex Liouville string. Additionally, we investigate the non-perturbative completion of the dual matrix model, matching the calculations of ZZ-instantons and uncovering non-perturbative properties of the double-scaled matrix model.

Furthermore, as we will see in section 3, a special feature of ZZ-instantons in the complex Liouville string is that their action is purely imaginary. As a result, ZZ-instantons mediate non-perturbative corrections to closed string amplitudes that are not exponentially suppressed, but rather oscillatory. This motivates us to also consider non-perturbative corrections mediated by ZZ-ghost-instantons, which have the same action but with the opposite sign [23, 24]. In other string theories, an oppositely signed action leads to ghost-instantons mediating non-perturbatively enhanced corrections to amplitudes, but in the complex Liouville string, these corrections remain oscillatory. In fact, we will see that swap symmetry — an important symmetry input in the bootstrap of perturbative amplitudes in [1] — implies the existence of ZZ-ghost-instantons in the non-perturbative sector.

**Non-perturbative completion of the dual matrix integral.** The contributions of ZZ-instantons on the worldsheet provide a window into the non-perturbative completion of the matrix integral dual of the complex Liouville string. We use matrix model techniques to deduce the leading non-perturbative corrections to the string amplitudes, and show that they are precisely accounted for by the ZZ-instanton contributions on the worldsheet. From these leading non-perturbative corrections together with resurgence techniques we then extract the large-genus asymptotics of the string amplitudes, which display the $(2g)!$ growth expected on

general grounds in string theory [25]. From these asymptotics we infer the effective string coupling, which we demonstrate to be *imaginary*. This indicates that the genus expansion of the string amplitudes exhibits sign oscillations. This can be traced back to the fact that the tension of the ZZ-instantons is purely imaginary, and relatedly, the corresponding non-perturbative effects give wildly oscillatory (rather than doubly-exponentially small) contributions to the string amplitudes.

These non-perturbative effects also have consequences for the eigenvalue spectrum and ultimately the convergence of the matrix integral. One notable feature of the eigenvalue density $\rho(E)$ of either of the two matrices of the two-matrix integral dual to the complex Liouville string [2] is its non-positivity in certain regions of the real energy axis. Initially, near the edge of the distribution, the density is positive, reaching a maximum before decreasing and eventually becoming negative. We will show that the point where the density of states vanishes corresponds precisely to the location of ZZ-instantons.

More specifically, the saddle points of the effective potential felt by a pair of eigenvalues (one from each matrix) in the presence of the others correspond to the nodal singularities of the spectral curve of the two-matrix integral. These, in turn, map directly to ZZ-instantons on the string worldsheet. Consequently, at the point where the density of states becomes negative, a new saddle begins to dominate the matrix integral (or, equivalently, the string field path integral on the string theory side), and requires a deformation of the integration contour for the eigenvalues into the complex plane.

Furthermore, in section 4 we will calculate the effects of these nodal singularities in the spectral curve and find precise agreement with the non-perturbative corrections to closed string amplitudes mediated by ZZ-instantons and ghost-ZZ-instantons.

**Outline of the paper.** The rest of the paper is organized as follows. In section 2.1 we give a general treatment of conformal boundary conditions for the worldsheet theory of the complex Liouville string and show how they may be used to compute more general observables, such as resolvents and partition functions, from the closed string amplitudes. This culminates in the gluing formula (2.32), which computes the perturbative contributions to such general observables by gluing suitable "trumpets" (corresponding to punctured disks) onto the closed string amplitudes. The gluing integral runs over the on-shell spectrum of the worldsheet theory. We discuss how the boundary conditions associated with matrix model quantities of interest arise from the more familiar Liouville BCFTs. In section 3 we put the formalism to work and compute the leading non-perturbative contributions of ZZ-instantons to the string amplitudes, in both the single- and multi-instanton sectors. We highlight the role of ghost instantons [23,24], which are needed for the swap symmetry of the non-perturbative string amplitudes. Finally in section 4 we discuss the consequences of these effects for the non-perturbative completion of the dual matrix model. We conclude with some discussion of open questions. We collect some background material and computations in the appendices.

## 2   Worldsheet boundaries

The natural observables of the complex Liouville string are the string amplitudes $\mathsf{A}_{g,n}^{(b)}(\boldsymbol{p})$, defined by integrating worldsheet conformal field theory correlation functions of closed string vertex operators over the moduli space of Riemann surfaces. These are however *not* directly the most natural observables from the point of view of the dual matrix integral [2] — those would be for example the resolvents or the partition functions of the matrix model. In order to discuss other observables, as well as ZZ-instantons corresponding to non-perturbative effects in the matrix model, we need to equip the worldsheet conformal field theory with conformal

boundary conditions. We will consider worldsheets with any number of boundaries labelled by possibly distinct boundary states $\{\Psi\}$ of the worldsheet theory. We denote the corresponding perturbative string amplitudes by

$$Z_{g,n}^{(b)}(\boldsymbol{\Psi}),\tag{2.1}$$

where we denote by $\boldsymbol{\Psi} = (\Psi_1, \dots, \Psi_n)$ a collection of boundary conditions and by $g$ the genus of the core of the worldsheet away from the conformal boundaries. As we shall see below, there is a special boundary condition for which the worldsheet boundary reduces back to a closed string vertex operator and thus these quantities in particular also contain the perturbative string amplitudes $A_{g,n}^{(b)}(\boldsymbol{p})$ as special cases. In that case we will slightly abuse notation and replace $\Psi_j$ by the corresponding Liouville momentum $p_j$. We use the letter $Z$ since we think of these quantities as partition functions.

There are some special cases of this general definition that will appear more frequently and deserve extra mention. For example, the disk partition function with boundary condition $\Psi$ will often be denoted by

$$Z_{\text{disk}}^{(b)}(\Psi) \equiv Z_{0,1}^{(b)}(\Psi),\tag{2.2}$$

while the so-called "trumpet" partition function is the disk partition function with an additional closed string vertex operator insertion and is denoted by

$$Z_{\text{trumpet}}^{(b)}(\Psi, p) \equiv Z_{0,2}^{(b)}(\Psi, p).\tag{2.3}$$

Here $p$ labels the Liouville momentum associated with the closed string vertex operator. The terminology is borrowed from models of two-dimensional dilaton gravity [6]. The relation to that language will be clarified in [4].

Since the standard conformal boundary conditions of Liouville CFT furnish a sufficiently complete basis for worldsheet BCFT observables, and conformal boundary conditions in non-unitary, non-compact worldsheet CFT are relatively weakly constrained anyhow, we will begin with a completely general discussion of conformal boundary conditions without committing to any particular family of worldsheet BCFT. This general treatment will lead us to a formulation of observables with worldsheet boundaries that may be interpreted as a smearing of closed string amplitudes. In this sense the closed string amplitudes contain all the information of these a priori more general boundary observables, and conformal boundary states that define certain observables may be interpreted in terms of superpositions of closed string vertex operators. We will then see how the standard observables such as resolvents and partition functions of the matrix model may be recast in this light.

## 2.1 General worldsheet boundaries

We start by considering a general conformal boundary state $|\Psi\rangle$ for the product of the two Liouville CFTs on the worldsheet.

**Conventions.** We usually decorate quantities for the first Liouville factor by a + superscript or no superscript if no confusion is possible and quantities for the second Liouville factor by a − superscript. We use the following standard parametrizations of the central charge and the conformal weights

$$c = 1 + 6Q^2, \qquad Q = b + b^{-1}, \qquad h_p = \tilde{h}_p = \frac{Q^2}{4} - p^2.\tag{2.4}$$

When $c \in 13 + i\mathbb{R}$, to guarantee cancellation of the Weyl anomaly of the worldsheet CFT the central charge of the second Liouvillle CFT is given by

$$c^- = 1 + 6(Q^-)^2, \qquad Q^- = (b^-)^{-1} + b^-, \qquad b^- = -ib, \qquad b \in e^{\frac{\pi i}{4}}\mathbb{R}_+.\tag{2.5}$$

The physical closed-string vertex operators of the $c^\pm$ Liouville CFTs furthermore need to obey the mass-shell condition leading to the following constraint when $h_p \in \frac{1}{2} + i\mathbb{R}$

$$h_p + h_{p^-} = 1 \quad \Leftrightarrow \quad p^- = ip, \qquad p \in e^{-\frac{\pi i}{4}}\mathbb{R}_+. \tag{2.6}$$

It is also convenient to define vertex operators with an additional leg-pole factor so that the full vertex operator reads $\mathcal{V}_p = \mathcal{N}_b(p)\,c\tilde{c}V_p^+ V_{ip}^-$ with $V_p^\pm$ the standard vertex operators of Liouville CFT. We denote the OPE measure in our conventions by $\rho_b(p) = 4\sqrt{2}\sin(2\pi bp)\sin(2\pi b^{-1}p)$. We refer to [1] for a complete description of our conventions.

**CFT boundary conditions.** Recall that conformal boundary conditions are characterized by a state in the Hilbert space of the CFT on the circle obtained by mapping the upper half-plane to the disk. Such a boundary state may be decomposed in terms of the Ishibashi states $|V_{p^\pm}^\pm\rangle\rangle$ associated with the primaries of the Liouville CFTs

$$|\Psi\rangle := -\int_0^{-i\infty} dp^+ \rho_b(p^+) \int_0^{-i\infty} dp^- \rho_{-ib}(p^-)\Psi(p^+, p^-)|V_{p^+}^+\rangle\rangle|V_{p^-}^-\rangle\rangle. \tag{2.7}$$

Here $\Psi(p^+, p^-)$ is a suitable function of the Liouville momenta that captures the one-point function of the primary $V_{p^+}^+ V_{p^-}^-$ of the combined worldsheet CFT on the disk. The contour of integration corresponds to the spectrum of Liouville CFT analytically continued to complex central charge. We will later discuss rotating the contour so that the internal spectrum is the one that is most natural in the complex Liouville string, namely $p \in e^{-\frac{\pi i}{4}}\mathbb{R}_+$, but for now we will retain the original spectrum of Liouville CFT. In principle the boundary state (2.7) need not factorize into each of its Liouville CFT constituents, though it will in the examples that we consider below. The Ishibashi states are normalized such that[2]

$$\langle\langle V_{p_1}^+|e^{-\pi t(L_0^+ + \tilde{L}_0^+ - \frac{c^+}{12})}|V_{p_2}^+\rangle\rangle = i\frac{\delta(p_1 - p_2) + \delta(p_1 + p_2)}{\rho_b(p_1)}\chi_{p_1}^{(b)}(it), \tag{2.8a}$$

$$\langle\langle V_{p_1}^-|e^{-\pi t(L_0^- + \tilde{L}_0^- - \frac{c^-}{12})}|V_{p_2}^-\rangle\rangle = i\frac{\delta(p_1 - p_2) + \delta(p_1 + p_2)}{\rho_{-ib}(p_1)}\chi_{p_1}^{(-ib)}(it). \tag{2.8b}$$

Here

$$\chi_p^{(b)}(it) = \frac{e^{2\pi tp^2}}{\eta(it)}, \tag{2.9}$$

is the non-degenerate Virasoro character. Here as usual we express the characters in terms of the Liouville momentum, related to the conformal weight as in (2.4). Hence the partition function of the combined Liouville CFTs on a cylinder of length $\pi t$ and unit radius with $\Psi$ and $\Phi$ conformal boundary conditions on either end is given by

$$Z_{0,2}^{(b)}(\Psi, \Phi; t) = -\int_0^{-i\infty} dp^+ \rho_b(p^+) \int_0^{-i\infty} dp^- \rho_{-ib}(p^-)\Psi(p^+, p^-)\Phi(p^+, p^-)\chi_{p^+}^{(b)}(it)\chi_{p^-}^{(-ib)}(it). \tag{2.10}$$

Familiar examples for the boundary state $|\Psi\rangle$ in the worldsheet CFT are combinations of the FZZT and the ZZ boundary conditions for the individual Liouville CFTs, which are reviewed in appendix A and will be employed later in section 2.3. In these cases, the cylinder modular bootstrap is a highly stringent constraint that demands an interpretation as a trace over a Hilbert space on the strip, decomposing the cylinder partition function into a sum of Virasoro

---

[2]The perhaps unfamiliar factor of $i$ in (2.8a) and (2.8b), as well as the overall minus sign in the boundary state (2.7), is an artifact of the change in conventions parameterizing the Liouville momenta in terms of $p = -iP$.

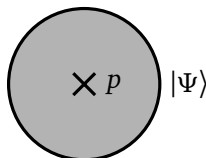

Figure 2: The one-point amplitude $Z^{(b)}_{\text{trumpet}}(\Psi, p)$ of the on-shell state labeled by the Liouville momentum $p$ on the disk with $\Psi$ conformal boundary conditions.

characters with positive coefficients in the dual channel. In the present more general discussion, it is not clear what role if any is played by unitarity of the boundary state (2.7), so for now we will be agnostic about positivity of the cylinder partition function in the dual channel. Indeed in string theory we only require positivity of the BRST cohomology of the combined worldsheet theory, and need not insist on unitarity of the constituent worldsheet CFTs in the usual CFT sense. In what follows we will however insist on a notion of normalizability of the boundary state that we will come to shortly.

**Disk and trumpet amplitudes.** The function $\Psi(p^+, p^-)$ that defines the boundary state encodes the disk one-point amplitude of the worldsheet BCFT defined by the boundary condition $\Psi$ (see figure 2). Indeed, the disk one-point amplitude is given by

$$Z^{(b)}_{\text{trumpet}}(\Psi, p) = \frac{C^{(b)}_{\text{D}^2}}{2\pi} \mathcal{N}_b(p) \Psi(p, ip). \tag{2.11}$$

Following conventions in JT gravity we will refer to the punctured disk amplitude as the "trumpet" partition function. Here we have used that on-shell $p^- = ip$. The prefactors correspond to the leg-pole factor that provides the normalization of the bulk vertex operator [1]

$$\mathcal{N}_b(p) = -\frac{(b^2 - b^{-2})\rho_b(p)}{8\sqrt{2}\pi p \sin(\pi b^2) \sin(\pi b^{-2})}, \tag{2.12}$$

and the normalization of the string theory path integral on the disk

$$C^{(b)}_{\text{D}^2} = C_{\text{D}^2} \frac{\sin(\pi b^2) \sin(\pi b^{-2})}{b^2 - b^{-2}}. \tag{2.13}$$

The latter normalization factor is proportional to the square-root of the normalization of the path integral on the sphere $C^{(b)}_{\text{S}^2}$ given in [1], up to the $b$-independent constant $C_{\text{D}^2}$ to be fixed below. The factor of $\frac{1}{2\pi}$ in (2.11) is due to the division of the residual conformal Killing group of a once-punctured disk, which is generated by rotations of the disk about the origin and whose volume is $2\pi$. Putting everything together we have

$$Z^{(b)}_{\text{trumpet}}(\Psi, p) = -\frac{C_{\text{D}^2}}{16\sqrt{2}\pi^2} \frac{\rho_b(p)}{p} \Psi(p, ip). \tag{2.14}$$

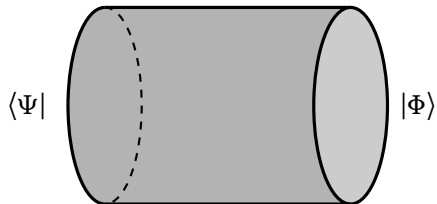

Figure 3: The cylinder amplitude $Z^{(b)}_{0,2}(\Psi, \Phi)$ between $\Psi$ and $\Phi$ conformal boundary conditions on each boundary of the cylinder.

Finally, the disk amplitude is computed as usual by applying the dilaton equation [2, eq. (4.25)] to the trumpet partition function:

$$Z_{\text{disk}}^{(b)}(\Psi) = b^{-1}\left[ Z_{\text{trumpet}}^{(b)}\big(\Psi, p = \tfrac{1}{2}(b + b^{-1})\big) + Z_{\text{trumpet}}^{(b)}\big(\Psi, p = \tfrac{1}{2}(b^{-1} - b)\big)\right]. \tag{2.15}$$

**Cylinder amplitude.** We now consider the amplitude of the complex Liouville string on the cylinder with conformal boundary conditions $\Psi$ and $\Phi$ on each end of the cylinder (see figure 3). It is computed by combining the cylinder partition function (2.10) with that of the $\mathfrak{bc}$-ghost system and integrating over the moduli space of the cylinder. This computation gives

$$Z_{0,2}^{(b)}(\Psi, \Phi) = \int_0^\infty \mathrm{d}t\, \eta(it)^2 Z_{0,2}^{(b)}(\Psi, \Phi; t), \tag{2.16}$$

assuming that $\Phi \neq \Psi$ and we are treating the two boundaries as distinguishable. In order to proceed we would like to swap the integral over the cylinder modulus $t$ with those over the Liouville momenta of the complete set of states propagating in the closed-string channel. Since $(p^+)^2, (p^-)^2 \in \mathbb{R}_-$ the $t$ integral is convergent and we have

$$Z_{0,2}^{(b)}(\Psi, \Phi) = \frac{1}{2\pi} \int_0^{-i\infty} \mathrm{d}p^+ \int_0^{-i\infty} \mathrm{d}p^- \frac{\rho_b(p^+)\rho_{-ib}(p^-)\Psi(p^+, p^-)\Phi(p^+, p^-)}{(p^+)^2 + (p^-)^2}. \tag{2.17}$$

Next, we deform the contour of integration over the Liouville momenta $p^-$ in order to pick up the residue located at $p^- = ip^+$. We recognize this pole as the location at which the intermediate bulk vertex operator precisely becomes on-shell. Under the assumption that the disk one-point functions $\Psi$ and $\Phi$ do not lead to additional singularities and are sufficiently well-behaved at infinity, we may perform this contour deformation and we obtain[3]

$$Z_{0,2}^{(b)}(\Psi, \Phi) = \frac{1}{4} \int_0^{-i\infty} \mathrm{d}p\, \frac{\rho_b(p)\rho_{-ib}(ip)\Psi(p, ip)\Phi(p, ip)}{p} \tag{2.18}$$

$$= \left(\frac{8\pi^2}{C_{\mathrm{D}^2}}\right)^2 \int_0^{-i\infty} (-2p\,\mathrm{d}p)\, Z_{\text{trumpet}}^{(b)}(\Psi, p)\, Z_{\text{trumpet}}^{(b)}(\Phi, p). \tag{2.19}$$

We refer to conformal boundary conditions obeying these properties as "normalizable" boundary conditions. Let us already mention that importantly this assumption turns out to be too optimistic for ZZ-instantons. Here we find it convenient to fix the overall normalization

$$C_{\mathrm{D}^2} = 8\pi^2. \tag{2.20}$$

A strong consistency check of this value will come from the precise matching of non-perturbative effects mediated by ZZ-instantons in section 3 as well as a factorization condition described later in this section and checked in appendix B. The end result is that the cylinder partition function is computed by integrating the trumpet partition functions (2.14) against the sphere two-point string amplitude $A_{0,2}^{(b)}$

$$Z_{0,2}^{(b)}(\Psi, \Phi) = \int_0^{-i\infty} (-2p\,\mathrm{d}p) \int_0^{-i\infty} (-2p'\,\mathrm{d}p')\, Z_{\text{trumpet}}^{(b)}(\Psi, p) Z_{\text{trumpet}}^{(b)}(\Phi, p') A_{0,2}^{(b)}(p, p'). \tag{2.21}$$

Recall that the sphere two-point amplitude on the half-line $p, p' \in i\mathbb{R}_-$ is given by [1]

$$A_{0,2}^{(b)}(p, p') = -\frac{1}{2p}\delta(p - p'). \tag{2.22}$$

---

[3]To get the full residue at $p^- = ip^+$ we extended contour of integration over $p^-$ to the full line $i\mathbb{R}$ and multiplied the result by $\frac{1}{2}$. This assumes that the that disk one-point functions $\Psi$ and $\Phi$ are even in $p^-$, which is the case for the examples of boundary conditions we consider in this paper.

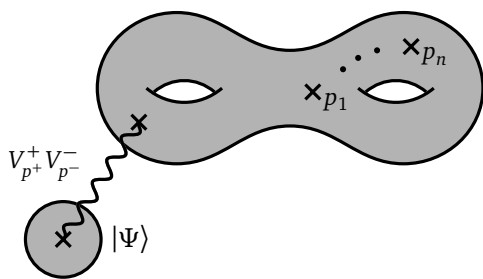

Figure 4: Gluing of a conformal boundary of type $\Psi$, represented by the disk with boundary condition $|\Psi\rangle$ on the lower left, to a string amplitude $\mathsf{A}^{(b)}_{g,n}(\boldsymbol{p})$, represented by the closed Riemann surface of genus $g$ and $n$ on-shell vertex operator insertions with momenta $p_1, \ldots, p_n$. The gluing is performed by summing over all off-shell states $V^+_{p^+} V^-_{p^-}$ with a propagator $q^{-(p^+)^2 - (p^-)^2}$ where $q = \mathrm{e}^{-2\pi t}$, represented by the wavy line, and integrating over the gluing modulus $t$.

**General gluing of conformal boundaries.** The previous example of the cylinder diagram can be further generalized to the gluing of conformal boundaries to a closed worldsheet diagram with genus $g$ and $n$ external on-shell string states. For instance, consider the addition of a single conformal boundary of the type (2.7) to a closed Riemann surface $\Sigma_{g,n}$ as pictured in figure 4. We use the plumbing fixture to perform the gluing of a half-infinite cylinder (which is conformally equivalent to a once-punctured disk) through a cylinder with gluing parameter $q = \mathrm{e}^{-2\pi t}$. The full moduli space to integrate over is the moduli space of surfaces with one boundary and $n$ closed string vertex operators. Via the plumbing fixture, it is isomorphic to $\mathcal{M}_{g,n+1} \times \mathbb{R}_+$, where $t$ parametrizes the $\mathbb{R}_+$ direction and the half-infinite cylinder gets attached to the $(n+1)^{\text{st}}$ vertex operator. Here it is important that there is no mapping class group mixing the two factors of $\mathcal{M}_{g,n+1} \times \mathbb{R}_+$.[4]

The resulting string amplitude may hence be written as[5]

$$Z^{(b)}_{g,n+1}(\Psi, \boldsymbol{p}) = C^{(b)}_{\Sigma_{g,1}} \int_{\mathcal{M}_{g,n+1} \times \mathbb{R}_+} \int \mathrm{d}p^+ \rho_b(p^+) \int \mathrm{d}p^- \rho_{-ib}(p^-) \Psi(p^+, p^-) q^{-(p^+)^2 - (p^-)^2}$$

$$\times \left\langle \mathcal{B}_t \, \mathcal{B}_0 \widetilde{\mathcal{B}}_0 \, c\tilde{c} \, V^+_{p^+} V^-_{p^-} \prod_{k=1}^{3g-3+n} \mathcal{B}_k \widetilde{\mathcal{B}}_k \prod_{j=1}^{n} \mathcal{V}_{p_j} \right\rangle_{\Sigma_{g,n+1}}, \tag{2.23}$$

where $q = \mathrm{e}^{-2\pi t}$ is a gluing parameter associated with the gluing of a cylinder with length $\pi$ and circumference $2\pi t$. Here, $\mathcal{B}_t$ is the $\flat$-ghost contour associated with the modulus $t$, $\mathcal{B}_0$ and $\widetilde{\mathcal{B}}_0$ are the ghost contours circling the vertex operator $V^+_p V^-_{p'}$, and the $\mathcal{B}_k$ are those associated to

---

[4]In contrast, recall that when factorizing a closed string amplitude into lower-point amplitudes using the plumbing fixture, we glue a closed string sub-diagram $\Sigma_{g_1,n_1}$ to another sub-diagram $\Sigma_{g_2,n_2}$ by attaching a cylinder with a moduli space parametrized by $C = \{\tau \in \mathbb{C} \,|\, 0 \leqslant \operatorname{Re}\tau \leqslant 2\pi, \operatorname{Im}\tau \geqslant 0\}$. Here, $g_1 + g_2 = g$ and $n_1 + n_2 = n + 2$, where $g$ and $n$ are the genus and number of on-shell closed string insertions of the original diagram. However, the associated moduli space of the plumbing fixture $\mathcal{M}_{g_1,n_1} \times \mathcal{M}_{g_2,n_2} \times C$ is not isomorphic to $\mathcal{M}_{g,n}$ as there is a non-trivial mapping class group mixing the different factors.

Put differently, when performing an OPE expansion near the degeneration limit of a closed string diagram, we cannot single out one OPE channel because when the gluing parameter $q$ becomes too large, we must expand in a dual channel. Thus, the moduli space of the gluing cylinder is not $C$, but a quotient thereof. In the case of gluing a half-infinite cylinder with boundary condition $\Psi$, however, there is no dual channel. This allows for the gluing of half-infinite cylinders similarly to gauge theory, where one only integrates over the monodromy of the gauge field around the stump of the amputated surface.

[5]In the state frame, (2.23) is obtained by evaluating the overlap between the boundary state (2.7) and the resolution of the identity $\mathbb{1} = -\int \mathrm{d}p^+ \rho_b(p^+) \int \mathrm{d}p^- \rho_{-ib}(p^-) |V^+_{p^+}\rangle |V^-_{p^-}\rangle \langle V^+_{p^+}| \langle V^-_{p^-}|$, where the minus sign is once again an artifact of our parametrization of the Liouville momenta in terms of $p = -iP$. Notice that this sign cancels with that of the boundary state. Evaluating this overlap leads to the operator expression (2.23).

the remaining moduli of the surface $\Sigma_{g,n}$. The $\mathcal{V}_{p_j} = \mathcal{N}_b(p_j) c\tilde{c} V^+_{p_j} V^-_{ip_j}$ are fixed on-shell vertex operators in the complex Liouville string, including the leg-pole factor (2.12). $C^{(b)}_{\Sigma_{g,m}}$ denotes the normalization constant of the string theory path integral on a Riemann surface with $g$ handles and $m$ boundaries; for instance, $C^{(b)}_{\Sigma_{g=0,m=1}} = C^{(b)}_{D^2}$. By $\langle \cdot \rangle$ we mean the correlation function in the full worldsheet CFT including ghosts. Recall also that the anticommuting objects $\mathcal{B}_t$, $\mathcal{B}_k$ and $\widetilde{\mathcal{B}}_k$ serve as the differentials in the integral over moduli space. In our conventions, we can identify $\mathcal{B}_t = \pi \, dt$.[6]

In principle, (2.23) contains subleading corrections in the gluing parameter $q$ coming from the subleading terms in the conformal block expansion. In the presence of ghosts, we can reduce the sum over descendants to the BRST cohomology of the worldsheet, since terms that are not in the BRST cohomology will cancel with their BRST partner after integrating over the remaining moduli of the correlator. However, in the complex Liouville string (and in similar 2d string theories), the cohomology is entirely localized on the primary states and thus all subleading corrections in the $q$-expansion are total derivatives on $\mathcal{M}_{g,n+1}$, which decouple. This argument is essentially identical to the observation that the discontinuities in the perturbative string amplitudes do not receive corrections from subleading terms in the OPE, which was discussed in [1].

The integrated worldsheet CFT correlator in (2.23) is not strictly speaking a string amplitude since the insertion $c\tilde{c} V^+_{p^+} V^-_{p^-}$ is not on-shell. However, as we did in the previous case of the cylinder diagram we can switch the order of integrations in (2.23) to obtain a string amplitude. Indeed, performing the integral over the modulus $t$, and deforming the $p^-$ contour to pick up the residue at $p^- = ip^+$, we obtain

$$
\begin{aligned}
Z^{(b)}_{g,n+1}(\Psi, \boldsymbol{p}) &= C^{(b)}_{\Sigma_{g,1}} \frac{1}{2} \int dp^+ \rho_b(p^+) \int dp^- \rho_{-ib}(p^-) \Psi(p^+, p^-) \frac{1}{(p^+)^2 + (p^-)^2} \\
&\quad \times \int_{\mathcal{M}_{g,n+1}} \left\langle \mathcal{B}_0 \widetilde{\mathcal{B}}_0 c\tilde{c} V^+_{p^+} V^-_{p^-} \prod_{k=1}^{3g-3+n} \mathcal{B}_k \widetilde{\mathcal{B}}_k \prod_{j=1}^{n} \mathcal{V}_{p_j} \right\rangle_{\Sigma_{g,n+1}} \\
&= C^{(b)}_{\Sigma_{g,1}} \frac{\pi}{8} \int (-2p \, dp) \left( \frac{\rho_b(p)}{p \, \mathcal{N}_b(p)} \right)^2 \frac{2\pi}{C^{(b)}_{D^2}} Z^{(b)}_{\text{trumpet}}(\Psi, p) \\
&\quad \times \int_{\mathcal{M}_{g,n+1}} \left\langle \mathcal{B}_0 \widetilde{\mathcal{B}}_0 \mathcal{N}_b(p) c\tilde{c} V^+_p V^-_{ip} \prod_{k=1}^{3g-3+n} \mathcal{B}_k \widetilde{\mathcal{B}}_k \prod_{j=1}^{n} \mathcal{V}_{p_j} \right\rangle_{\Sigma_{g,n+1}},
\end{aligned}
\tag{2.25}
$$

where we used (2.11) to write the disk one-point function $\Psi(p, ip)$ in terms of $Z^{(b)}_{\text{trumpet}}(\Psi, p)$. We dropped the superscript $+$ on the $p^+$ integral. Since the vertex operator $V^+_p V^-_{ip}$ is now on-shell, we can write [1]

$$
C^{(b)}_{\Sigma_g} \int_{\mathcal{M}_{g,n+1}} \left\langle \mathcal{B}_0 \widetilde{\mathcal{B}}_0 \mathcal{N}_b(p) c\tilde{c} V^+_p V^-_{ip} \prod_{k=1}^{3g-3+n} \mathcal{B}_k \widetilde{\mathcal{B}}_k \prod_{j=1}^{n} \mathcal{V}_{p_j} \right\rangle_{\Sigma_{g,n+1}} = A^{(b)}_{g,n+1}(p, \boldsymbol{p}). \tag{2.26}
$$

---

[6]Our conventions for the $\mathfrak{b}$-ghost contour $\mathcal{B}_t$ are as follows. Parameterizing the cylinder by $0 \leqslant \text{Re} \, w \leqslant \pi$ and $w \sim w + 2\pi t$ [26], $\mathcal{B}_t$ can be taken to be (excluding the differential $dt$)

$$
\mathcal{B}_t = \int_L \left( \mathfrak{b}(w) dw + \tilde{\mathfrak{b}}(\bar{w}) d\bar{w} \right), \tag{2.24}
$$

where $L$ is a horizontal segment from $\text{Re} \, w = 0$ to $\text{Re} \, w = \pi$. Expanding $\mathfrak{b}(w)$ into modes we obtain that $\mathcal{B}_t \to \pi \mathfrak{b}_0^+$ with $\mathfrak{b}_0^+ = \mathfrak{b}_0 + \tilde{\mathfrak{b}}_0$.

Therefore, using (2.26) we can write (2.25) as

$$Z_{g,n+1}^{(b)}(\Psi, \boldsymbol{p}) = \frac{C_{\Sigma_{g,1}}^{(b)}}{C_{\Sigma_g}^{(b)} C_{D^2}^{(b)}} \frac{\pi^2}{4} \int (-2p \, \mathrm{d}p) \left(\frac{\rho_b(p)}{p \, \mathcal{N}_b(p)}\right)^2 Z_{\text{trumpet}}^{(b)}(\Psi, p) A_{g,n+1}^{(b)}(p, \boldsymbol{p}). \quad (2.27)$$

Notice that the combination in the parenthesis squared in (2.27) is actually $p$-independent,

$$\frac{\rho_b(p)}{p \, \mathcal{N}_b(p)} = -8\sqrt{2}\pi \, \frac{\sin(\pi b^2)\sin(\pi b^{-2})}{b^2 - b^{-2}}, \quad (2.28)$$

where we used (2.12). Thus, we obtain that

$$Z_{g,n+1}^{(b)}(\Psi, \boldsymbol{p}) = \left[\frac{32\pi^4 C_{\Sigma_{g,1}}^{(b)}}{C_{\Sigma_g}^{(b)} C_{D^2}^{(b)}} \left(\frac{\sin(\pi b^2)\sin(\pi b^{-2})}{b^2 - b^{-2}}\right)^2\right] \int (-2p \, \mathrm{d}p) \, Z_{\text{trumpet}}^{(b)}(\Psi, p) A_{g,n+1}^{(b)}(p, \boldsymbol{p}).$$
$$(2.29)$$

The prefactor in the square brackets is expected to be precisely unity

$$\frac{32\pi^4 C_{\Sigma_{g,1}}^{(b)}}{C_{\Sigma_g}^{(b)} C_{D^2}^{(b)}} \left(\frac{\sin(\pi b^2)\sin(\pi b^{-2})}{b^2 - b^{-2}}\right)^2 \overset{!}{=} 1. \quad (2.30)$$

In fact, the argument presented in this paragraph is the same as the analysis of factorization of perturbative string amplitudes in worldsheet string theory [26]. In particular, this analysis is one way in which the normalization constants $C_{\Sigma_{g,m}}$ may be fixed indirectly. Appendix B provides a check of the condition (2.30) for the case of gluing one boundary to the sphere $n$-point function.

Assuming the condition (2.30), we finally obtain the general result that we may add conformal boundaries with boundary condition $\Psi$ to any string amplitude $A_{g,n}^{(b)}$ by gluing a trumpet partition function with a gluing measure $(-2p \, \mathrm{d}p)$:

$$Z_{g,n+1}^{(b)}(\Psi, \boldsymbol{p}) = \int_0^{-i\infty} (-2p \, \mathrm{d}p) \, Z_{\text{trumpet}}^{(b)}(\Psi, p) A_{g,n+1}^{(b)}(p, \boldsymbol{p}). \quad (2.31)$$

For example, if we glue $n$ conformal boundaries to a worldsheet string diagram without any additional external on-shell vertex operator insertions, we obtain an observable that we may think of as analogous to partition functions with $n$ asymptotic boundaries in models of 2d dilaton gravity

$$Z_{g,n}^{(b)}(\Psi_1, \ldots, \Psi_n) = \int_0^{-i\infty} \prod_{j=1}^{n} \left(-2p_j \, \mathrm{d}p_j \, Z_{\text{trumpet}}^{(b)}(\Psi_j, p_j)\right) A_{g,n}^{(b)}(p_1, \ldots, p_n). \quad (2.32)$$

This observable is characterized by the $n$ boundary conditions $\{\Psi_j\}$ that we glue to the closed string insertions. This presentation makes it clear that we may interpret worldsheet boundaries in terms of smeared closed-string vertex operators. Indeed, observables with worldsheet boundaries are simply computed by inserting a complete set of on-shell states and integrating the closed string amplitude against the trumpet partition functions corresponding to the conformal boundaries with the measure endowed by the two-point function in target space, see figure 1. That the smearing of the bulk vertex operators commutes with the integral over moduli space is ensured by the normalizability of the boundary condition.

In the case of the "trivial boundary condition," which we denote by a Liouville momentum (say, $q$) corresponding to an ordinary closed-string puncture, the "trumpet partition function" is simply given by the sphere two-point amplitude itself[7]

$$Z_{\text{trumpet}}^{(b)}(q,p) = A_{0,2}^{(b)}(p,q) = -\frac{\delta(p-q)}{2p}. \tag{2.33}$$

**Seiberg-Shih equivalence.** We see that we only need to specify the trumpet partition function $Z_{\text{trumpet}}^{(b)}(\Psi,p)$ to compute arbitrary observables with the boundary condition $\Psi$. The trumpet partition function (2.14) in turn only requires the wave function for on-shell vertex operators, $\Psi(p,ip)$. In particular there can be different boundary conditions $\Psi$ and $\Phi$ which satisfy $\Psi(p,ip) = \Phi(p,ip)$ and thus are equivalent for the purpose of (on-shell) string theory. This means that a given boundary condition can be realized in multiple ways in the worldsheet theory. One can think of this phenomenon as the worldsheet BRST cohomology reducing the number of possible boundary conditions. This was first explained in the context of the minimal string by Seiberg and Shih in [11] and we call it the Seiberg-Shih equivalence.

## 2.2 Conformal boundary conditions for physical observables

We now explore a few families of conformal boundary conditions that will be natural in various physical contexts. They may all be straightforwardly related to each other by suitable integral transforms.

### 2.2.1 Resolvent

The natural observables of the two-matrix integral dual of the complex Liouville string [2] are resolvents of one of the two matrices. Recall that this matrix integral is characterized by the spectral curve

$$x(w) = -2\cos(\pi b^{-1}w), \qquad y(w) = 2\cos(\pi bw). \tag{2.34}$$

Here $w$ is a convenient coordinate on the spectral curve. Before double-scaling, the resolvent associated with the first matrix $M_1$ is defined by the trace

$$R(x) = \text{tr}\,\frac{1}{x - M_1}. \tag{2.35}$$

In the large-$N$ limit, the connected part of products of resolvents admits a genus expansion

$$\left\langle \prod_{j=1}^{n} R(x_j) \right\rangle_{\text{c}} = \sum_{g=0}^{\infty} R_{g,n}(x_1,\ldots,x_n)N^{2-2g-n}. \tag{2.36}$$

Upon double-scaling, a natural family of observables is furnished by the resolvent differentials

$$\omega_{g,n}^{(b)}(w_1,\ldots,w_n) = R_{g,n}(x(w_1),\ldots,x(w_n))\,dx(w_1)\cdots dx(w_n). \tag{2.37}$$

The resolvent differentials are entirely determined by the spectral curve (2.34) by topological recursion, see e.g. [2,27]. In (2.34) we have picked a specific parameterization of the spectral curve. The resolvents associated with other parameterizations are of course just related by Jacobian factors.

---

[7]In what follows we will denote the trumpet partition functions by a parameter (such as $q$ below) that characterizes the conformal boundary state, whose meaning should be clear from the context.

We claim that the perturbative contributions to the resolvents $\omega_{g,n}^{(b)}$ are captured by the following trumpet partition function

$$Z_{\text{trumpet}}^{(b)}(w,p) = -2\pi \sin(2\pi w p)\,dw\,. \tag{2.38}$$

Notice that we have included the differential $dw$ in the definition of the trumpet partition function. The boundary conditions for the resolvents then form a complete basis for general conformal boundary conditions, since the trumpet partition function of any normalizable boundary condition may be decomposed into those of the resolvents by standard Fourier analysis.[8]

In particular, the general resolvents are then given by the following integral transform of the closed string amplitudes, similarly to how one glues trumpets to the Weil-Petersson volumes to introduce asymptotic boundaries in JT gravity [6][9]

$$
\begin{aligned}
Z_{g,n}^{(b)}(w_1,\ldots,w_n) &= \int_\gamma \left( \prod_{j=1}^n (-2p_j\,dp_j)\, Z_{\text{trumpet}}^{(b)}(w_j,p_j) \right) A_{g,n}^{(b)}(p_1,\ldots,p_n) \\
&= \omega_{g,n}^{(b)}(w_1,\ldots,w_n)\,.
\end{aligned}
\tag{2.39}
$$

Indeed, with the trumpet partition function given by (2.38), this is precisely the relation between the string amplitudes and the resolvent differentials explained in our previous paper [2].

So we conclude that the matrix model resolvents may be thought of as closed string amplitudes computed with suitably smeared vertex operators, or alternatively in terms of worldsheets equipped with appropriate conformal boundary conditions whose punctured disk amplitudes are given by (2.38).[10] In particular the cylinder amplitude correctly reproduces the leading contribution to the two-point resolvent in this parameterization of the spectral curve [2]

$$Z_{0,2}^{(b)}(w_1,w_2) = dw_1\,dw_2\,(2\pi)^2 \int_\gamma (-2p\,dp)\,\sin(2\pi w_1 p)\sin(2\pi w_2 p) \tag{2.40}$$

$$= dw_1\,dw_2 \left( \frac{1}{(w_1-w_2)^2} - \frac{1}{(w_1+w_2)^2} \right) \tag{2.41}$$

$$= \omega_{0,2}^{(b)}(w_1,w_2)\,. \tag{2.42}$$

As explained in footnote 9, to obtain the second line, we expanded the trigonometric functions into exponentials and declared that $\int_\gamma dp\,p\,e^{ap} = a^{-2}$ for arbitrary complex $a$, by choosing the contour to run in an appropriate directions so that the integral converges.

### 2.2.2 Thermal partition function

Another observable that one might wish to consider from the matrix integral description of the complex Liouville string are thermal partition functions. In single-matrix integrals where one may interpret the matrix as the Hamiltonian of a quantum system the thermal partition functions are very natural observables. But in a two-matrix integral like the dual of the complex

---

[8]The general trumpet partition function must be an odd function of the Liouville momentum $p$ due to the non-standard leg-pole factor (2.12) that defines the normalization of closed string vertex operators.

[9]Here we are avoiding committing to a specific contour; in practice the integral is computed by expanding the products of trigonometric functions that appear into complex exponentials and deforming the contour (originally a vertical half-line in the $p_j$ plane) in an appropriate direction such that the integral converges. This is what the abstract contour $\gamma$ is meant to represent.

[10]Operationally, in section 2.3 we will see that the resolvent boundary conditions are simply related to conformal boundary conditions of the type FZZT$(u)\times$ZZ$_{(1,1)}$, where each factor corresponds to each Liouville component of the matter worldsheet CFT. The standard Liouville BCFTs are reviewed in appendix A.

Liouville string the definition is less canonical and one has to make a specific choice. For example, one could define a thermal partition function by the following trace involving the first matrix $M_1$

$$Z(\beta) = \operatorname{tr} e^{-\beta M_1}. \tag{2.43}$$

In the double-scaling limit the connected expectation value of products of these partition functions admit a genus expansion similarly to the resolvents.

   We claim that the trumpet amplitudes corresponding to the thermal partition functions in the complex Liouville string are given by the following expression

$$Z_{\text{trumpet}}^{(b)}(\beta, p) = \frac{2b}{\pi} \sin(2\pi b p) K_{2bp}(2\beta), \tag{2.44}$$

where $K_\nu$ is the modified Bessel function of the second kind. This may be obtained from the resolvent trumpet partition function (2.38) by inverse Laplace transform with respect to $x = -2\cos(\pi b^{-1} w)$, which follows from the relation between (2.35) and (2.43) and the parameterization of the spectral curve (2.34). We will discuss this in more detail in section 2.3, where we will see that these boundary conditions may moreover be obtained in terms of the more familiar conformal boundaries for Liouville CFT by combining FZZT boundary conditions for one worldsheet Liouville CFT with ZZ boundary conditions for the other.

   Indeed, the disk amplitude that one computes from (2.44) via the dilaton equation precisely reproduces the corresponding thermal partition function in the matrix integral

$$Z_{\text{disk}}^{(b)}(\beta) = b^{-1} \left[ Z_{\text{trumpet}}^{(b)}\big(\beta, p = \tfrac{1}{2}(b^{-1} + b)\big) + Z_{\text{trumpet}}^{(b)}\big(\beta, p = \tfrac{1}{2}(b^{-1} - b)\big) \right] \tag{2.45}$$

$$= -\frac{2ib^2}{\pi\beta} \sinh(-\pi i b^2) K_{b^2}(2\beta). \tag{2.46}$$

The disk partition function may equivalently be recast as the following integral over the spectrum of the first matrix

$$Z_{\text{disk}}^{(b)}(\beta) = -\frac{8 \sinh(-\pi i b^2)}{b} \int_0^{e^{-\frac{\pi i}{4}} \infty} \mathrm{d}u \, e^{-2\beta \cos(2\pi b^{-1} u)} \sin(2\pi b u) \sin(2\pi b^{-1} u) \tag{2.47}$$

$$= \int_2^\infty \mathrm{d}E \, e^{-\beta E} \rho_0(E), \tag{2.48}$$

where $\rho_0(E)$ is the leading density of eigenvalues of the first matrix extracted from the spectral curve (2.34) [2]

$$\rho_0(E) = \frac{2}{\pi} \sinh(-\pi i b^2) \sin\left(-ib^2 \operatorname{arccosh}\left(\frac{E}{2}\right)\right). \tag{2.49}$$

One might worry that in (2.48) we are integrating the leading density of eigenvalues (2.49) over a region where it becomes non-sign definite. We will address this when we discuss the non-perturbative completion of the matrix integral in section 4.

   The perturbative expansion of products of thermal partition functions are then computed by gluing trumpets (2.44) onto the corresponding closed string amplitudes as in (2.32).

**Duality symmetry and a second partition function.**   The worldsheet theory of the complex Liouville string exhibits "duality symmetry", which may be stated as the following invariance of the string amplitudes

$$\mathsf{A}_{g,n}^{(b^{-1})}(\boldsymbol{p}) = (-1)^n \mathsf{A}_{g,n}^{(b)}(\boldsymbol{p}). \tag{2.50}$$

This property is manifest from the definition of the worldsheet theory as the $b \to b^{-1}$ symmetry of Liouville CFT, but is highly nontrivial from the point of view of the dual matrix integral. It

follows from what is known as $x$-$y$ symmetry of the matrix model, which amounts to the statement that the free energies may equivalently be computed from the topological recursion with the roles of $x$ and $y$ in the spectral curve (2.34) exchanged [28, 29]. The boundary state corresponding to the thermal partition functions as defined by (2.43) clearly breaks this symmetry (see for instance the trumpet amplitude (2.44)), which is due to the fact that the definition of the partition function treats the two matrices asymmetrically. This fact can be taken as an indication that the dual matrix model has to be in fact a two-matrix model to describe the boundary conditions completely, see [30, 31] for similar comments.

There is correspondingly another natural partition function that one may wish to consider, which involves a trace over the second matrix rather than the first

$$\widetilde{Z}(\beta) = \mathrm{tr}\, e^{-\beta M_2}\,. \tag{2.51}$$

In this case, the trumpet amplitude is simply the image of the original trumpet amplitudes (2.44) under the action of the duality symmetry $b \to b^{-1}$

$$\widetilde{Z}^{(b)}_{\mathrm{trumpet}}(\beta, p) = Z^{(b^{-1})}_{\mathrm{trumpet}}(\beta, p) = \frac{2}{\pi b}\sin(2\pi b^{-1}p)K_{2b^{-1}p}(2\beta)\,. \tag{2.52}$$

With this in hand we conclude that the general perturbative contribution to the product of partition functions associated with the second matrix (2.51) is given in terms of the image of that associated with the first matrix under the duality symmetry up to a sign

$$\widetilde{Z}^{(b)}_{g,n}(\beta_1,\dots,\beta_n) = \int_\gamma \left(\prod_{j=1}^n (-2p_j\,\mathrm{d}p_j)\widetilde{Z}^{(b)}_{\mathrm{trumpet}}(\beta_j, p_j)\right) A^{(b)}_{g,n}(p_1,\dots,p_n) \tag{2.53}$$

$$= (-1)^n Z^{(b^{-1})}_{g,n}(\beta_1,\dots,\beta_n)\,. \tag{2.54}$$

### 2.2.3 ZZ-instantons

Another very physically-relevant family of conformal boundaries are the ZZ-instanton boundary conditions, which we will make extensive use of in section 3. At the level of the worldsheet BCFT, they are defined by combining $(r,s)$ ZZ boundary conditions for one Liouville CFT with $(1,1)$ ZZ boundary conditions for the other Liouville CFT. We review the BCFT associated with these conformal boundary conditions in appendix A. We will hence label the combined boundary conditions by the pair of positive integers $(r,s)$. The ZZ boundary state defines Dirichlet-like boundary conditions for Liouville CFT so for each $(r,s)$ this defines a species of localized instantons in the combined worldsheet theory. The trumpet partition function is computed by combining the ZZ disk one-point functions (A.3) together with the leg-pole factor and normalization of the path integral on the once-punctured disk

$$Z^{(b)}_{\mathrm{trumpet}}((r,s), p) = \frac{C^{(b)}_{\mathrm{D^2}}}{2\pi}\mathcal{N}_b(p)\psi^{(b)}_{(r,s)}(p)\psi^{(-ib)}_{(1,1)}(ip) = \frac{2\sin(2\pi r bp)\sin(2\pi s b^{-1}p)}{p}\,, \tag{2.55}$$

where we used (2.13) and (2.20). Here in a slight abuse of notation we write $\Psi = (r,s)$ to refer to the corresponding ZZ-instanton boundary condition. This family of conformal boundary conditions is not strictly normalizable in the sense described in the previous subsection; $b^{-1}p$ is purely imaginary so the partition function grows exponentially at large $p$. However we will see in what follows that we may nevertheless treat them similarly to how we have treated normalizable boundary conditions thus far, at least for some simple quantities. As we will mention in the discussion section 5, this breaks down for more complicated quantities. This is because the string theoretic moduli space integrals for the ZZ-instantons don't converge

and one has to change the contour to properly define them. In some cases, this is correctly accounted for by using the general formula (2.32).

We may proceed to compute the cylinder partition function by analytic continuation. In practice, we expand the product of sines from (2.55) that appear in (2.32) and for each term deform the contour of integration in a direction such that the integral converges. In this way we find

$$
\begin{aligned}
Z_{0,2}^{(b)}((r,s),(r',s')) &= \int_\gamma (-2p\,\mathrm{d}p)\,Z_{\text{trumpet}}^{(b)}((r,s),p)\,Z_{\text{trumpet}}^{(b)}((r',s'),p) \\
&= \log\left[\frac{((r-r')^2 b^2 - (s-s')^2 b^{-2})((r+r')^2 b^2 - (s+s')^2 b^{-2})}{((r-r')^2 b^2 - (s+s')^2 b^{-2})((r+r')^2 b^2 - (s-s')^2 b^{-2})}\right].
\end{aligned}
\tag{2.56}
$$

We will see in section 3 that this agrees with the more direct worldsheet computation. The case with $(r,s) = (r',s')$ naively gives a divergent answer. In that case the above computation does not apply and one requires string field theory to treat it properly.

## 2.3 Worldsheet boundaries from Liouville BCFT

Here we will explore worldsheet boundaries in the language of the more familiar conformal boundary conditions for Liouville CFT. We will see that the boundary conditions discussed in the previous subsection may be constructed by combining the basic conformal boundary states of FZZT($u$) and ZZ$_{(r,s)}$ type, that were reviewed in appendix A, from each Liouville component of the full worldsheet CFT. The main nontrivial example is the one that results from combing an FZZT($u$) boundary state in the first Liouville component together with a ZZ$_{(1,1)}$ boundary state in the second Liouville component. This type of conformal boundary in the complex Liouville string is the one that most closely resembles the kind of boundary conditions that appropriately introduce boundaries of finite length in the $(p,q)$ minimal string [11,14,32,33].

**FZZT($u$)×ZZ conformal boundary conditions.** We will mainly discuss the worldsheet boundary conformal field theory that results from equipping one worldsheet Liouville CFT with FZZT boundary conditions and the other with $(1,1)$ ZZ boundary conditions. In particular, we will consider the worldsheet boundary state

$$
|\text{FZZT}^{(b)}(u)\rangle \otimes |\text{ZZ}_{(1,1)}^{(-ib)}\rangle\,.
\tag{2.57}
$$

This boundary state provides a complete basis to describe the normalizable boundary conditions of the previous subsection. $u$ can in principle be any complex number. The trumpet partition function associated with this conformal boundary condition is given by

$$
Z_{\text{trumpet}}^{(b)}(u,p) = \frac{C_{\mathrm{D}^2}^{(b)}}{2\pi}\mathcal{N}_b(p)\psi^{(b)}(u;p)\psi_{(1,1)}^{(-ib)}(ip) = \frac{\cos(4\pi up)}{p}\,.
\tag{2.58}
$$

We will see in what follows how the normalizable conformal boundary conditions discussed in the previous subsection may be obtained from simple manipulations of this basic boundary condition.

**Seiberg-Shih equivalence.** In (2.57) we chose the basis $(1,1)$ ZZ boundary state for the second Liouville theory. By the Seiberg-Shih equivalence discussed above, we can always choose one of the boundary conditions to be the $(1,1)$ ZZ boundary state. We could alternatively also construct such a complete basis by considering $(r,s)$ ZZ boundary states.

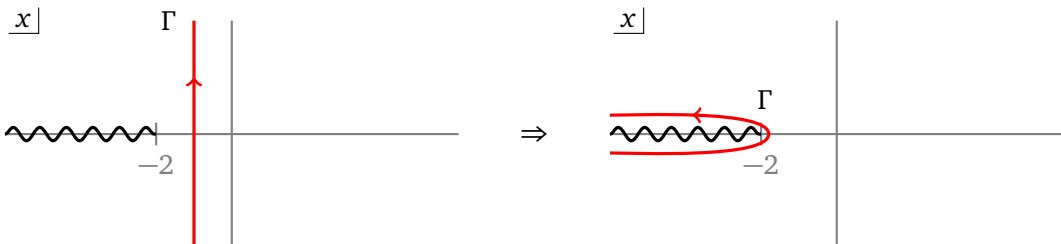

Figure 5: The partition function of the matrix model is related to the resolvent by an inverse Laplace transform. In computing the trumpet amplitude for the thermal partition functions we evaluate the inverse Laplace transform by wrapping the contour $\Gamma$ around the branch cut running from $x = -2$ to $x = -\infty$.

**Relation to resolvent boundary conditions.** The conformal boundary conditions that define the resolvents of the matrix integral dual of the complex Liouville string that were discussed in section 2.2.1 are very simply related to the FZZT×ZZ worldsheet boundary conformal field theory. Comparing the trumpet partition functions (2.38) and (2.58) we see that

$$|\omega(w)\rangle = \partial_w \left( |\text{FZZT}^{(b)}(u = \tfrac{w}{2})\rangle \otimes |\text{ZZ}_{(1,1)}^{(-ib)}\rangle \right) dw\,. \tag{2.59}$$

So we conclude that the resolvent boundary conditions are related to the more familiar FZZT×ZZ boundary conditions by a simple derivative with respect to the FZZT parameter $u$.

**Relation to thermal partition function boundary conditions.** Here we will see that the FZZT×ZZ boundary conditions are also simply related to the thermal partition function boundary conditions discussed in section 2.2.2, although the relation is more involved than for the resolvents. This relation elucidates the boundary conditions used to describe thermal partition functions via Liouville BCFT in the previous literature on the $(p,q)$ minimal string and Liouville gravity [11, 14, 33]. We know from (2.35) and (2.43) that the partition functions are obtained from the resolvents by an inverse Laplace transform with respect to the $x$ parameter of the spectral curve (2.34). Combining this with the relation between the boundary states that defines the resolvents and the FZZT×ZZ conformal boundaries (2.59), we conclude that the boundary state that defines the thermal partition functions is given by[11]

$$\begin{aligned}
|Z(\beta)\rangle &= \frac{1}{2\pi i} \int_\Gamma e^{\beta x} |\omega(w = \tfrac{b}{\pi} \arccos(\tfrac{x}{2}))\rangle \\
&= \frac{1}{2\pi i} \int_\Gamma dx\, e^{\beta x} \partial_x \left( |\text{FZZT}^{(b)}(u = \tfrac{b}{2\pi} \arccos(\tfrac{x}{2}))\rangle \otimes |\text{ZZ}_{(1,1)}^{(-ib)}\rangle \right).
\end{aligned} \tag{2.60}$$

Here $\Gamma$ is a contour that runs vertically to the right of the singularities of the integrand in the $x$ plane. In previous literature on Liouville gravity and the minimal string, $x$ was roughly identified with the boundary cosmological constant characterizing the FZZT boundary conditions and the derivative with respect to $x$ was interpreted as the insertion of a "marking operator" that ungauges translations along the boundary circle [14, 34]. The inverse Laplace transform was then interpreted as a change of ensemble from fixed boundary cosmological constant to fixed boundary length. Here we have essentially derived this procedure starting from the relationship between the resolvent and the thermal partition function in the matrix model. We

---

[11]More concretely, here we are using that $Z(\beta) = -\frac{1}{2\pi i} \int_\Gamma dx\, e^{\beta x} R(-x)$ together with the spectral curve relation $w = \frac{b}{\pi} \arccos(-\tfrac{x}{2})$ from (2.34).

could equivalently have integrated by parts to write

$$|Z(\beta)\rangle = -\frac{\beta}{2\pi i} \int_\Gamma dx\, e^{\beta x}\, |\text{FZZT}^{(b)}(u = \tfrac{b}{2\pi}\arccos(\tfrac{x}{2}))\rangle \otimes |ZZ^{(-ib)}_{(1,1)}\rangle\,, \qquad (2.61)$$

where the factor of $\beta$ is interpreted as a result of the "marking" of the boundary circle.

For example, we can straightforwardly evaluate the trumpet partition function this way. Recalling the expression (2.58) for the trumpet partition function associated with the FZZT×ZZ boundary conditions and plugging into (2.60), we have[12]

$$Z^{(b)}_{\text{trumpet}}(\beta, p) = \frac{b}{2\pi i} \int_\Gamma dx\, e^{\beta x} \frac{\sin(2bp\arccos(\tfrac{x}{2}))}{\sin(\arccos(\tfrac{x}{2}))} \qquad (2.62)$$

$$= -\frac{2b\sin(2\pi bp)}{2\pi i} \int_2^\infty dE\, e^{-\beta E} \frac{\cos(2bp\arccos(\tfrac{E}{2}))}{\sin(\arccos(\tfrac{E}{2}))}\,. \qquad (2.63)$$

In the second line we wrapped the contour around the branch cut running from $x = -2$ to $x = -\infty$ (see figure 5) and evaluated the discontinuity across the cut using

$$\arccos(z \pm i\varepsilon) = \mp\arccos(|z|) + \pi\,, \qquad z < -1\,. \qquad (2.64)$$

We can simplify this further by recognizing the integral representation of the modified Bessel function of the second kind to arrive at

$$Z^{(b)}_{\text{trumpet}}(\beta, p) = \frac{2b}{\pi}\sin(2\pi bp)K_{2bp}(2\beta)\,, \qquad (2.65)$$

which is the expression we anticipated in section 2.2.2. It in particular leads to a disk amplitude (2.46) that is consistent with the leading density of eigenvalues of the first matrix (2.49) extracted from the spectral curve (2.34).

# 3 ZZ-instantons

We now move on to study boundary conditions corresponding to instanton corrections in the genus expansion of the amplitudes $A^{(b)}_n(S_0; \boldsymbol{p})$. These cannot quite be treated in the way that we explained so far, as there are certain divergences that need to be treated with the help of string field theory. Given their importance for the non-perturbative structure of the theory, we discuss their evaluation in detail.

## 3.1 Generalities

The existence of non-perturbative contributions of order $e^{\#g_s^{-1}}$ to closed string scattering amplitudes has long been anticipated and understood to originate from D-instanton effects [25, 35, 36]. Recently, this class of non-perturbative effects has been studied and calculated more systematically, at leading and sub-leading orders in $g_s$ about a given instanton sector, in low-dimensional string theories [13, 16–22, 37–44] and in higher-dimensional superstrings [45–48]. In many cases, these calculations of D-instanton corrections to closed string scattering have led to a precise agreement with, when available, independent determination of such non-perturbative corrections from holographic and/or stringy dualities. In particular, it has been

---

[12]In order for the integral to be convergent, we strictly speaking have to assume that $|bp| < \frac{1}{2}$ in this computation. The result for larger values of $bp$ can for example be obtained by analytic continuation, by interpreting the integral distributionally, or by slightly bending the contour $\Gamma$ to the left.

recently recognized [18, 22, 37, 38, 48] that string field theory is crucial both to understand the origin of divergences and ambiguities present in a naive on-shell worldsheet D-instanton calculus as well as to resolve them in a systematic fashion.

Additionally, a priori one might worry that this class of non-perturbative corrections are ambiguous without a proper understanding of the resummation of the closed string perturbative genus expansion of scattering amplitudes. Indeed, recent calculations of D-instanton corrections have been performed in theories for which the perturbative genus expansion resummation is explicitly known from a dual description [13, 16–18, 20–22, 42–44], or when the non-perturbative correction is due to a D-instanton configuration of minimal charge or that is otherwise unambiguous [36, 45, 46, 48–51]. Low-dimensional string theory examples have the virtue that the resurgence properties can be studied in detail from the dual matrix integral perspective. From the matrix side, a choice is made for the contour of integration over the eigenvalues of the matrix integral. On the string theory side this translates to a choice of contour of integration in the path integral over open string fields living on the D-instanton [20, 21]. In fact, in these cases there is very strong evidence that the full open-string perturbative expansion around a given D-instanton sector can indeed be matched between the matrix integral and the string theory descriptions. That is, specific Lefschetz thimbles (as opposed to simply a contour of integration passing through a particular linear combination of saddle points) can be matched between the two dual descriptions.

## 3.2 One-instanton sector

**Boundary conditions.** D-instantons or ZZ-instantons in the complex Liouville string are defined on the worldsheet through a product of $(r, s)$ ZZ boundary conditions for each Liouville CFT component. However, as in the case of the $(p, q)$ or the Virasoro minimal string, a complete basis for the ZZ-instanton boundary conditions, at the level of the full string theory, is given by the product of $(r, s)$ ZZ boundary condition and the $(1, 1)$ ZZ boundary condition in each Liouville component, respectively [11],

$$|ZZ^{(b)}_{(r,s)}\rangle \otimes |ZZ^{(-ib)}_{(1,1)}\rangle \,. \tag{3.1}$$

We think of these as an irreducible basis for the more general boundary conditions that involve pairing $(k, \ell)$ ZZ boundary conditions for one Liouville factor with $(m, n)$ ZZ boundary conditions for the other. For example, we have

$$Z^{(b)}_{\text{trumpet}}(((k,\ell),(m,n)),p) = \sum_{r=|k-m|+1}^{k+m-1} \sum_{s=|\ell-n|+1}^{\ell+n-1} Z^{(b)}_{\text{trumpet}}(((r,s),(1,1)),p)\,. \tag{3.2}$$

In what follows we will leave implicit the $(1, 1)$ label and refer to (3.1) as $(r, s)$ ZZ-instanton boundary conditions.

In this section, we will be interested in the leading-order correction to the string amplitudes $A^{(b)}_n(S_0; \boldsymbol{p})$ mediated by a single $(r, s)$ ZZ-instanton. We denote this correction as $A^{(b)}_n(S_0; \boldsymbol{p})^{(r,s)_1}_{\pm}$, where the superscript $(r, s)_1$ indicates the type $(r, s)$ ZZ-instanton, with its multiplicity given in the subscript. In this section, the multiplicity is set to one for a single instanton, though more general multiplicities $\ell_{r,s} \in \mathbb{Z}_{\geqslant 1}$ will be considered later. The subscript $\pm$ corresponds to the instanton and ghost-instanton contributions, respectively, which we will explain further below.

**Leading diagram.** In the on-shell formalism, this contribution is captured by the following worldsheet diagram,

$$A_n(S_0; \boldsymbol{p})^{(r,s)_1}_{+} = \exp\left(e^{S_0}\,\bullet + \bigcirc\right)\,\otimes\cdots\otimes + \cdots, \tag{3.3}$$

where we inserted appropriate factors of $g_s = e^{-S_0}$. Here we impose the same boundary condition (3.1) on all worldsheet boundaries.

In fact, ZZ-instanton corrections admits an *open*-string genus expansion as a series in all integer powers of $g_s = e^{-S_0}$. The $+\cdots$ in (3.3) include higher topologies that are suppressed by factors of $e^{-S_0}$. In this paper, we will focus on the leading-order contribution given in (3.3) and, hopefully without room for confusion, will omit the $+\cdots$ in $A_n(S_0; \boldsymbol{p})_+^{(r,s)_1}$ in the rest of this section.

**Disk.** The disk one-point diagram for the case of ZZ-instanton boundary conditions was already introduced in (2.55), repeated here for convenience,

$$\otimes \equiv Z_{\text{trumpet}}^{(b)}((r,s),p) = \frac{C_{\text{D}^2}^{(b)}}{2\pi} \mathcal{N}_b(p)\, \psi_{(r,s)}^{(b)}(p)\, \psi_{(1,1)}^{(-ib)}(p) = \frac{2\sin(2\pi r b p)\sin(2\pi s b^{-1} p)}{p}. \quad (3.4)$$

The empty disk diagram is related to the action, or tension, of the $(r,s)$ ZZ-instanton. The exponentiated empty disk diagram gives rise to the characteristic non-perturbative dependence $e^{\# g_s^{-1}}$. This may be readily obtained by applying the dilaton equation (2.15) to the disk one-point diagram

$$\bigcirc \equiv Z_{\text{disk}}^{(b)}(r,s) = b^{-1}\left[ Z_{\text{trumpet}}^{(b)}\left((r,s),p = \tfrac{b^{-1}+b}{2}\right) + Z_{\text{trumpet}}^{(b)}\left((r,s),p = \tfrac{b^{-1}-b}{2}\right) \right]$$
$$= -8(-1)^{r+s}\frac{\sin(\pi r b^2)\sin(\pi s b^{-2})}{b^{-2}-b^2}. \quad (3.5)$$

The tension of the ZZ-instanton is minus the disk partition function with the appropriate $g_s$-dependence,

$$T_{r,s}^{(b)} = -e^{S_0} \bigcirc = 8e^{S_0}(-1)^{r+s}\frac{\sin(\pi r b^2)\sin(\pi s b^{-2})}{b^{-2}-b^2}. \quad (3.6)$$

This value for the tension $T_{r,s}^{(b)}$ of the $(r,s)$ ZZ-instanton is in precise agreement with the matrix model computation (4.4) presented in the following section.[13] Importantly, the tension (3.6) is purely imaginary for purely imaginary $b^2$. Thus, the non-perturbative corrections in this model are actually rapidly oscillating functions of order 1, instead of exponentially decaying. We will have more to say about this in section 4.

**Cylinder.** The cylinder diagram with the same $(r,s)$ ZZ-instanton boundary condition (3.1) on both ends of the cylinder gives rise to a normalization constant. We will evaluate it from scratch, since as we already remarked, we need extra string field theory input to make sense of it. These extra subtleties manifested themselves in the divergence of the general cylinder diagram (2.56). This worldsheet cylinder diagram can be evaluated as an integral over the modulus of a cylinder with circumference $\pi$ and length $2\pi t$ of the full open string spectrum,

$$\bigcirc \equiv Z_{0,2}^{(b)}((r,s),(r,s)) = \int_0^\infty \frac{dt}{2t}\, \eta(it)^2\, \chi_{(1,1)}^{(-ib)}(it) \sum_{r'=1}^{2r-1} \sum_{s'=1}^{2s-1} \chi_{(r',s')}^{(b)}(it)$$
$$= \sum_{r'=1}^{2r-1} \sum_{s'=1}^{2s-1} \int_0^\infty \frac{dt}{2t}(1-e^{-2\pi t})(1-e^{-2\pi rst})\, e^{\frac{\pi t}{2}((r'-1)b+(s'+1)b^{-1})((r'+1)b+(s'-1)b^{-1})}. \quad (3.7)$$

In the first line, the $\eta(it)^2$ is the contribution from the $\mathfrak{bc}$ ghosts, $\chi_{(1,1)}^{(-ib)}(it)$ is the spectrum between identical $(1,1)$ ZZ boundary states described in (A.2), and the rest is the spectrum

---

[13]Note that this matching provides a strong consistency check of the value of $C_{\text{D}^2}$ determined in (2.20).

between identical $(r,s)$ ZZ boundary states described in (A.4). The factor of $\frac{1}{2}$ accounts for the $\mathbb{Z}_2$ symmetry that exchanges the two boundaries of the cylinder.[14] This moduli integral over the cylinder length $t$ is convergent in the closed-string channel $t \to 0$, but suffers from an open-string divergence in the $t \to \infty$ limit coming from the $(r',s') = (1,1)$ term in the sum over characters. Let us separate this $(r',s') = (1,1)$ term in the cylinder diagram,

$$
\begin{aligned}
Z_{0,2}^{(b)}((r,s),(r,s)) = \sum_{\substack{r'\overset{2}{=}1 \\ (r',s')\neq(1,1)}}^{2r-1} \sum_{s'\overset{2}{=}1}^{2s-1} \int_0^\infty \frac{\mathrm{d}t}{2t}(1-e^{-2\pi t})(1-e^{-2\pi rs't'}) \\
\times e^{\frac{\pi t}{2}((r'-1)b+(s'+1)b^{-1})((r'+1)b+(s'-1)b^{-1})} + \int_0^\infty \frac{\mathrm{d}t}{2t}\left(e^{-2\pi t}+e^{2\pi t}-2\right).
\end{aligned}
\tag{3.8}
$$

More generally, the on-shell worldsheet D-instanton calculus, via diagrams of the form in (3.3), appears to be incomplete due to finite ambiguities in the D-instanton corrected amplitudes coming from divergences that appear near degeneration limits of moduli space. These divergences arise from several sources.

The first kind of divergences arise from the collective coordinates of the D-instanton, which correspond to zero modes (or, open string fields with a vanishing propagator) and that give rise to logarithmic divergences in the integral over moduli space of Riemann surfaces. These divergences are expected to cancel after summing over all relevant worldsheet diagrams with distinct topologies [16, 35, 36], but a finite additive constant ambiguity in the amplitude remains and that is only known to be unambiguously determined in the framework of string field theory [18, 37]. In the complex Liouville string, however, ZZ-instantons defined by (3.1) do not carry any collective coordinate, and thus this type of divergence will not be relevant.

A second source of divergences are the presence of tachyonic open strings on the D-instanton. In the context of D-instantons, these tachyonic modes do not signal the breakdown of the quantum theory, but instead are interpreted as a saddle point of the (open plus closed string field) action that give a sensible contribution to the full path integral and hence to scattering amplitudes. The contributions from tachyonic modes may be evaluated by analytic continuation.[15] For instance, the contribution from the $(r',s')^{\text{th}}$ term in the first line of (3.8) to the cylinder amplitude of a single ZZ-instanton can be evaluated as

$$
\begin{aligned}
\int_0^\infty \frac{\mathrm{d}t}{2t}(1-e^{-2\pi t})(1-e^{-2\pi r's't})\, e^{\frac{\pi t}{2}((r'-1)b+(s'+1)b^{-1})((r'+1)b+(s'-1)b^{-1})} \\
= \frac{1}{2}\log\left[\frac{((r'-1)b\pm(s'-1)b^{-1})((r'+1)b\pm(s'+1)b^{-1})}{((r'-1)b\pm(s'+1)b^{-1})((r'+1)b\pm(s'-1)b^{-1})}\right].
\end{aligned}
\tag{3.9}
$$

Here we take the product over both choices of signs on the RHS. (3.9) holds assuming that the integral converges, $\mathrm{Re}[((r'-1)b+(s'+1)b^{-1})((r'+1)b+(s'-1)b^{-1})] < 0$. In fact, for the branch of interest of the Liouville parameter $b = e^{\frac{\pi i}{4}}\mathbb{R}_+$, this is never the case. However, we can take (3.9) to be valid even when the exponent is such that the integral is divergent, i.e. corresponding to a tachyonic mode.[16] A second example of a divergence coming from a tachyonic open string mode on the ZZ-instanton is the term $e^{2\pi t}$ inside the parenthesis in the second line of (3.8).

---

[14] Note that the cylinder diagram (3.7) is the same as the cylinder diagram in VMS with ZZ-instanton boundary conditions after taking $b \to ib$, apart from the *finite* sum over degenerate characters in the dual channel labeled by $(r,s)$ in the present case.

[15] In the string field theory framework, tachyonic modes do not give rise to divergences since the worldsheet representation of its propagator $\int_0^\infty \mathrm{d}s\, e^{-m^2 s}$ is simply $\frac{1}{m^2}$, even for $m^2 < 0$.

[16] Note that here for $b = e^{\frac{\pi i}{4}}\mathbb{R}_+$, the argument of the logarithm on the RHS of (3.9) is always real and positive, unlike in the VMS where it takes both positive or negative values.

The third source of divergences, similar to the first mentioned above, are open string zero modes that however are not associated to the collective coordinates of the D-instanton. In the present case, these zero modes appear as the $-2$ term of the second contribution to the cylinder diagram in (3.8). This type of zero modes are much more subtle but have been recently understood using insights from string field theory, resulting in fully unambiguous D-instanton amplitudes in different theories [18, 20, 22, 38, 48]. These modes have a formally infinite propagator, that is a vanishing kinetic term in the string field action, but cannot be resolved by treating them as collective coordinates to be removed and integrated over at the end of the calculation. These modes, often referred to as ghost zero modes, instead signal the breakdown of the familiar Siegel gauge-fixing condition. As proposed by Sen [18, 38], one can instead employ a different gauge-fixing condition in the zero mode sector. This leads to additional out-of-Siegel-gauge zero modes that need to be included in the computation of the cylinder diagram.

Furthermore, the deformation of the contour of integration for the tachyonic modes in complexified field space introduces an ambiguity in selecting the full contour of integration, which determines a specific linear combination of Lefschetz thimbles. This ambiguity is expected to have a counterpart in the dual matrix model, where a similar ambiguity arises in the choice of contour for the eigenvalues of the two-matrix model. Since the theory does not a priori dictate the correct choice of contour, we remain agnostic and introduce a Stokes constant, denoted $\mathcal{S}_+^{(r,s)_1}$, to parameterize the multiplicity of the saddle in string field space. The subscript 1 refers to the single-instanton case, which will be generalized later.

This precise string field theory analysis can be found in [18, 20, 38, 48]. Here, we will simply apply the result of this procedure in the conventions of [20] in which the choice $\mathcal{S}_+^{(r,s)_1} = \frac{1}{2}$ was made, which leads to the interpretation

$$\exp\left[\int_0^\infty \frac{\mathrm{d}t}{2t}\left(\mathrm{e}^{-2\pi t} + \mathrm{e}^{2\pi t} - 2\right)\right] \longrightarrow \frac{i\,\mathcal{S}_+^{(r,s)_1}}{(2\pi)^{\frac{3}{2}}(T_{r,s}^{(b)})^{\frac{1}{2}}}, \tag{3.10}$$

where $T_{r,s}^{(b)}$ is the $(r,s)$ ZZ-instanton action, given by (3.6) in the complex Liouville string. The Stokes constant $\mathcal{S}_+^{(r,s)_1}$ constitutes non-perturbative information that goes beyond the cylinder diagram. Therefore, we will not include it directly in the cylinder diagram (3.7) but instead add it as an overall constant in the full ZZ-instanton contribution (3.3) (cf. (3.12) below).

Finally, combining (3.9) and (3.10) we obtain that the exponentiated cylinder diagram is given by

$$\exp\left(Z_{0,2}^{(b)}((r,s),(r,s))\right) = \frac{i}{(2\pi)^{\frac{3}{2}}(T_{r,s}^{(b)})^{\frac{1}{2}}}$$
$$\times \prod_{\substack{r'=1 \\ (r',s')\neq(1,1)}}^{2r-1}\prod_{s'=1}^{2s-1}\left[\frac{((r'-1)b \pm (s'-1)b^{-1})((r'+1)b \pm (s'+1)b^{-1})}{((r'-1)b \pm (s'+1)b^{-1})((r'+1)b \pm (s'-1)b^{-1})}\right]^{\frac{1}{2}}$$
$$= \frac{i}{(2\pi)^{\frac{3}{2}}(T_{r,s}^{(b)})^{\frac{1}{2}}}\left(\frac{\frac{b^{-2}}{r^2} - \frac{b^2}{s^2}}{b^{-2} - b^2}\right)^{\frac{1}{2}}. \tag{3.11}$$

**Summary.** By plugging these ingredients into (3.3), we obtain that the leading order single $(r,s)$ ZZ-instanton correction to the $n$-point amplitude in complex Liouville string theory is given by

$$A_n^{(b)}(S_0; \boldsymbol{p})_+^{(r,s)_1} = \frac{i\,\mathcal{S}_+^{(r,s)_1}\,\mathrm{e}^{-T_{r,s}^{(b)}}}{(2\pi)^{\frac{3}{2}}(T_{r,s}^{(b)})^{\frac{1}{2}}}\left(\frac{\frac{b^{-2}}{r^2} - \frac{b^2}{s^2}}{b^{-2} - b^2}\right)^{\frac{1}{2}}\prod_{j=1}^n \frac{2\sin(2\pi r b p_j)\sin(2\pi s b^{-1}p_j)}{p_j}, \tag{3.12}$$

where the tension $T_{r,s}^{(b)}$ of the ZZ-instanton is given by (3.6) and we have included the Stokes constant $S_+^{(r,s)_1}$. Let us note that we have been somewhat liberal in the choice of square root. In the final expression (3.12), we choose by convention the principal branch. Taking another branch could be absorbed in the definition of the Stokes constant $S_+^{(r,s)_1}$.

### 3.3 Ghost instantons

Let us explain one important issue. Looking back at the definition of the BCFT disk one-point functions $\psi_{(r,s)}^{(b)}(p)$, we could have equally taken $-\psi_{(r,s)}^{(b)}(p)$ as the disk one-point functions as mentioned below (A.3). Indeed, the cylinder bootstrap constraint that determines them is quadratic and unchanged under the inclusion of an overall minus sign. Thus there is a second boundary state which differs from (A.1) by an overall minus sign,

$$-|ZZ_{(r,s)}^{(b)}\rangle \otimes |ZZ_{(1,1)}^{(-ib)}\rangle \,. \tag{3.13}$$

Repeating the same computation as above with this boundary condition is easy as we just have to decorate a diagram with $n$ boundaries by $(-1)^n$. For the cylinder diagram, some care is needed as the string field theory computation also makes use of the tension of the instanton which *does* get modified for the ghost instanton. We thus get the second non-perturbative correction,

$$A_n^{(b)}(S_0;\boldsymbol{p})_-^{(r,s)_1} = \frac{i\,S_-^{(r,s)_1}\,e^{T_{r,s}^{(b)}}(-1)^n}{(2\pi)^{\frac{3}{2}}\big(-T_{r,s}^{(b)}\big)^{\frac{1}{2}}} \left(\frac{\frac{b^{-2}}{r^2}-\frac{b^2}{s^2}}{b^{-2}-b^2}\right)^{\frac{1}{2}} \prod_{j=1}^n \frac{2\sin(2\pi r b p_j)\sin(2\pi s b^{-1}p_j)}{p_j}\,. \tag{3.14}$$

In particular, the tension is opposite. For a real tension, this would mean that this contribution is exponentially enhanced and would dominate over the perturbative contributions. Thus such ghost instantons usually only appear once we analytically continue the coupling $S_0$ of the theory [23, 52]. In this model the tensions are however purely imaginary and hence the two non-perturbative corrections are of the same magnitude.

**Swap symmetry.** There is in fact an easy way to see that we need both these contributions. For this we recall that all perturbative contributions are invariant under swapping the two factors in the definition of the worldsheet theory. This amounts to the following swap symmetry [1]

$$A_{g,n}^{(-ib)}(i\boldsymbol{p}) = (-i)^n A_{g,n}^{(b)}(\boldsymbol{p})\,. \tag{3.15}$$

This symmetry should continue to hold also in the non-perturbative sector. The swap symmetry maps the instanton correction to the ghost instanton correction,

$$
\begin{aligned}
A_n^{(-ib)}(S_0;i\boldsymbol{p})_+^{(r,s)_1} &= \frac{i\,S_+^{(r,s)_1}\,e^{T_{r,s}^{(b)}}i^n}{(2\pi)^{\frac{3}{2}}\big(-T_{r,s}^{(b)}\big)^{\frac{1}{2}}} \left(\frac{\frac{b^{-2}}{r^2}-\frac{b^2}{s^2}}{b^{-2}-b^2}\right)^{\frac{1}{2}} \prod_{j=1}^n \frac{2\sin(2\pi r b p_j)\sin(2\pi s b^{-1}p_j)}{p_j} \\
&= (-i)^n \frac{S_+^{(r,s)_1}}{S_-^{(r,s)_1}} A_n^{(b)}(S_0;\boldsymbol{p})_-^{(r,s)_1}\,.
\end{aligned}
\tag{3.16}
$$

Thus insisting on the non-perturbative extension of the swap symmetry means that we need both contributions with equal Stokes constants,

$$S^{(r,s)_1} \equiv S_+^{(r,s)_1} = S_-^{(r,s)_1}\,. \tag{3.17}$$

### 3.4 Multi instantons

We can also consider leading-order non-perturbative contributions to $A_{g,n}^{(b)}(\boldsymbol{p})$ from a configuration of multi-instantons or of multi-ghost-instantons.[17] For example, the worldsheet diagram that computes the leading-order two-instanton contribution to the $n$-point string amplitude $A_{g,n}^{(b)}(\boldsymbol{p})$ is given by

$$\exp\left(e^{S_0}\underset{}{\bigcirc}^1 + \underset{}{\textcircled{1}}^1 + e^{S_0}\underset{}{\bigcirc}^2 + \underset{}{\textcircled{2}}^2 + \underset{}{\textcircled{2}}^1\right)\left(\underset{}{\otimes}^1 + \underset{}{\otimes}^2\right)\cdots\left(\underset{}{\otimes}^1 + \underset{}{\otimes}^2\right). \quad (3.18)$$

The boundaries labeled by 1 correspond to the first ZZ-instanton, say of type $(r,s)$, while those labeled by 2 correspond to the second ZZ-instanton, say of type $(r',s') \neq (r,s)$. In the second line of (3.18), the crosses in the first set of parenthesis correspond to an on-shell vertex operator with Liouville momentum $p_1$, the crosses in the second set of parenthesis are on-shell operators with momentum $p_2$, etc. The combinatorics of such contributions and of the cylinder diagrams involved can be taken into account by considering the Chan-Paton factors associated to the open string degrees of freedom on the ZZ-instantons.

**Identical instantons.** The new ingredients in a multi-instanton configuration in the complex Liouville string are the following. Consider first a configuration of $\ell \in \mathbb{Z}_{\geqslant 0}$ identical ZZ-instantons of type $(r,s)$. The associated action/tension of a multi-intanton configuration of $\ell$ ZZ-instantons is simply $\ell \times T_{r,s}^{(b)}$. Therefore, such a multi-instanton configuration provides a correction that is suppressed (or enhanced, in the ghost instanton case) by a factor of $e^{-\ell T_{r,s}^{(b)}}$; as noted earlier, in the complex Liouville string the tension (3.6) is purely imaginary and instanton corrections are oscillatory instead of suppressed. Second, for a system of $\ell$ identical ZZ-instantons, the open string spectrum is repeated $\ell^2$ times, and therefore we may simply raise the single-instanton empty cylinder diagram to the power $\ell^2$. The third and most nontrivial new ingredient is the division of the $U(\ell)$ gauge group on the ZZ-instanton worldvolume theory. This volume was computed and recently applied to the string-theoretic computation of multi-instanton corrections in type IIB string theory [46] and in the $(p,q)$ minimal string [21]. The net result for the non-perturbative correction to the partition function (i.e. the string amplitude without external on-shell states), for example, due to a configuration of identical $\ell$ ZZ-instantons (or ghost-instantons) can be written as [21, eq. 2.23]

$$A_0^{(b)}(S_0)_{\pm}^{(r,s)_\ell} = \mathcal{S}_{\pm}^{(r,s)_\ell} e^{\mp \ell T_{r,s}^{(b)}}\left[2\pi \exp\left(Z_{0,2}^{(b)}((r,s),(r,s))\right)\right]^{\ell^2}\frac{G_2(\ell+1)}{(2\pi)^{\frac{1}{2}\ell(\ell+1)}}, \quad (3.19)$$

where $G_2(x)$ is the Barnes-G double gamma function. The last factor in (3.19) is the reciprocal of $\mathrm{Vol}(U(\ell))$. It appears because there is a $U(\ell)$ gauge symmetry on the instanton worldvolume which needs to be divided by. This also leads to the extra $2\pi$ in the cylinder diagram which arises from the volume of $U(1)$ and which was already included in (3.10). The upper signs correspond to a multi-instanton contribution and the lower signs to the analogous multi-ghost-instanton contribution. Here, we have included the overall Stokes constant $\mathcal{S}_{\pm}^{(r,s)_\ell}$ associated with the $(r,s)_\ell$-instanton (or -ghost-instanton) sector.[18] Finally, including insertions of external

---

[17]We restrict our attention to a system of either all regular instantons or all ghost instantons. We leave the study of multi-instanton effects from a system with both instanton and ghost instantons to future work.

[18]Here we consider the generic situation in which each multi-instanton sector with $\ell$ ZZ-instantons carries its own Stokes constant. See [53–55], however, for recent examples in the context of topological strings which provide strong evidence that in those theories the Stokes constants are in fact independent of the multiplicity $\ell$ of a system of identical instantons. In this paper we will not attempt to explicitly compute these Stokes constants.

on-shell strings, we can write the leading order non-perturbative correction to the $n$-point amplitude $A_{g,n}^{(b)}(\boldsymbol{p})$ as

$$A_n^{(b)}(S_0; \boldsymbol{p})_{\pm}^{(r,s)_\ell} = S_{\pm}^{(r,s)_\ell} \, e^{\mp \ell \, T_{r,s}^{(b)}} \Big[ 2\pi \exp\big(Z_{0,2}^{(b)}((r,s),(r,s))\big) \Big]^{\ell^2} \frac{G_2(\ell+1)}{(2\pi)^{\frac{1}{2}\ell(\ell+1)}}$$
$$\times \ell^n \prod_{j=1}^n \Big( \pm Z_{0,1}^{(b)}((r,s),p_j) \Big). \tag{3.20}$$

**Several distinct instantons.**   We may further generalize to a system comprising several distinct ZZ-instantons (or ghost-instantons), indexed by a set $\mathcal{I} = \{(r,s)_{\ell_{r,s}}\}$. Here, the subscript $\ell_{r,s} \in \mathbb{Z}_{\geqslant 1}$ denotes the multiplicity of each type $(r,s)$ of ZZ-instanton included in the system. In this case, the leading order non-perturbative correction to the string amplitude $A_n^{(b)}(S_0; \boldsymbol{p})$ due to the multi-instanton system of ZZ-instantons $\mathcal{I}$ is given by

$$A_n^{(b)}(S_0; \boldsymbol{p})_{\pm}^{\mathcal{I}} = S_{\pm}^{\mathcal{I}} \left\{ \prod_{(r,s) \in \mathcal{I}} e^{\mp \ell_{r,s} \, T_{r,s}^{(b)}} \Big[ 2\pi \exp\big(Z_{0,2}^{(b)}((r,s),(r,s))\big) \Big]^{(\ell_{r,s})^2} \frac{G_2(\ell_{r,s}+1)}{(2\pi)^{\frac{1}{2}\ell_{r,s}(\ell_{r,s}+1)}} \right\}$$
$$\times \left\{ \prod_{\substack{(r,s),(r',s') \in \mathcal{I} \\ (r,s) < (r',s')}} \big( \exp Z_{0,2}^{(b)}((r,s),(r',s')) \big)^{\ell_{r,s} \ell_{r',s'}} \right\} \tag{3.21}$$
$$\times \left\{ \prod_{j=1}^n \sum_{(r,s) \in \mathcal{I}} \ell_{r,s} \big( \pm Z_{0,1}^{(b)}((r,s),p_j) \big) \right\},$$

where $Z_{0,2}^{(b)}((r,s),(r',s'))$ is the exponentiated cylinder diagram with distinct ZZ-instanton boundary conditions $(r,s)$ and $(r',s')$ on each end. Again, the power $\ell_{r,s} \times \ell_{r',s'}$ to which this cylinder diagram is raised simply accounts for the Chan-Paton factors of the open strings stretched between the two ZZ-instantons. We have also included the Stokes constant $S_{\pm}^{\mathcal{I}}$ associated to the multi-instanton configuration specified by the set $\mathcal{I}$. Notice that (3.18) is a special case of (3.21) with $\ell_{r,s} = 1$, $\ell_{r',s'} = 1$ and all other $\ell$'s vanishing.

Therefore, the only new ingredient in the worldsheet diagrammatics is the cylinder diagram $Z_{0,2}^{(b)}((r,s),(r',s'))$ with distinct ZZ-boundary conditions for each end. In the complex Liouville string, it can be computed as[19]

$$Z_{0,2}^{(b)}((r,s),(r',s')) = \int_0^\infty \frac{dt}{t} \, \eta(it)^2 \chi_{(1,1)}^{(-ib)}(it) \sum_{r'' \overset{2}{=} |r-r'|+1}^{r+r'-1} \sum_{s'' \overset{2}{=} |s-s'|+1}^{s+s'-1} \chi_{(r'',s'')}^{(b)}(it)$$
$$= \sum_{r'' \overset{2}{=} |r-r'|+1}^{r+r'-1} \sum_{s'' \overset{2}{=} |s-s'|+1}^{s+s'-1} \int_0^\infty \frac{dt}{2t} \big(1 - e^{-2\pi t}\big) \big(1 - e^{-2\pi r s'' t''}\big) \tag{3.22}$$
$$\times e^{\frac{\pi t}{2}((r''-1)b+(s''+1)b^{-1})((r''+1)b+(s''-1)b^{-1})}.$$

In contrast to (3.7), in this case we do not encounter divergences arising from open string zero modes (assuming $(r,s) \neq (r',s')$) and the cylinder diagram can be evaluated as was done in

---

[19]In this case the measure of integration over the modulus $t$ of the cylinder is $\frac{dt}{t}$, instead of $\frac{dt}{2t}$, because the two ends of the cylinder are distinct.

(3.9), with the result

$$Z_{0,2}^{(b)}((r,s),(r',s')) = \log\left[\prod_{r''\overset{2}{=}|r-r'|+1}^{r+r'-1}\prod_{s''\overset{2}{=}|s-s'|+1}^{s+s'-1}\frac{((r''-1)b\pm(s''-1)b^{-1})((r''+1)b\pm(s''+1)b^{-1})}{((r''-1)b\pm(s''+1)b^{-1})((r''+1)b\pm(s''-1)b^{-1})}\right]$$

$$= \log\left[\frac{((r-r')^2b^2-(s-s')^2b^{-2})((r+r')^2b^2-(s+s')^2b^{-2})}{((r-r')^2b^2-(s+s')^2b^{-2})((r+r')^2b^2-(s-s')^2b^{-2})}\right]. \tag{3.23}$$

As noted in section 2.2, this cylinder diagram is just a specific example of gluing two "trumpet" partition functions with ZZ-instanton boundary conditions, given by (3.4), with the measure $-2p\,\mathrm{d}p$. Indeed, we see that (3.23) is exactly the same as (2.56).

As we will see in the next section, this class of ZZ-instanton corrections is in precise agreement with the dual matrix model, providing a highly nontrivial and non-perturbative test of the duality.

**Stokes constants and swap symmetry.** We can easily generalize the argument leading to (3.17) to the multi-instanton case. $Z_{0,2}^{(b)}((r,s),(r',s'))$ given in (3.23) is invariant under swap symmetry, while the cylinder amplitude with identical boundaries $Z_{0,2}^{(b)}((r,s),(r,s))$ transforms correctly into its ghost counterpart. The origin of the factor $(-i)^n$ comes entirely from the disk one-point diagrams entering (3.21). Thus imposing swap symmetry on the multi-instanton corrections gives

$$\mathcal{S}^{\mathcal{I}} \equiv \mathcal{S}_+^{\mathcal{I}} = \mathcal{S}_-^{\mathcal{I}}. \tag{3.24}$$

# 4 Non-perturbative completion of the matrix model

We now discuss the non-perturbative definition of the matrix integral. This is a subtle issue since it is double scaled. Thus to define a non-perturbative completion, we have to first go away from the double scaling limit, derive a formula for a non-perturbative contribution and then take the double scaling limit. We will be content with discussing the leading non-perturbative corrections. From the point of view of the gravitational path integral, these correspond to 'doubly non-perturbative effects' [6]. They are particularly interesting since they probe the discreteness of the matrix model spectrum, but are difficult to access directly from the gravitational path integral.[20]

## 4.1 Leading non-perturbative effects

**Nodal singularities on the spectral curve.** The discussion of individual non-perturbative effects is largely analogous to the minimal string [11,12]. They are associated to nodal singularities in the spectral curve, which takes the form

$$\mathsf{x}(z) = -2\cos(\pi b^{-1}\sqrt{z}), \qquad \mathsf{y}(z) = 2\cos(\pi b\sqrt{z}). \tag{4.1}$$

Thus, we expect one species of ZZ-instanton for every nodal singularity of the spectral curve. They are labelled by $r,s \in \mathbb{Z}_{\geqslant 1}$, which indeed maps to the ZZ-instanton label $(r,s)$ in the bulk. As explained in [2], nodal singularities are located at

$$z_{(r,s)}^{\pm} = (rb\pm sb^{-1})^2, \tag{4.2}$$

where the surface intersects itself, since $z_{(r,s)}^+$ and $z_{(r,s)}^-$ maps to the same point under $(\mathsf{x}(z),\mathsf{y}(z))$.

---

[20]See also [56–59] for recent non-perturbative studies of matrix integrals dual to JT gravity, minimal strings, and Virasoro minimal strings.

**Partition function.**   Let us first discuss the partition function $A_0^{(b)}(S_0)$. The leading non-perturbative correction due to a single nodal singularity of the spectral curve labelled by the pair of integers $(r,s)$ can be written as [21, 60–62]

$$A_0^{(b)}(S_0)_\pm^{(r,s)_1} = \frac{1}{2\pi} \mathcal{B}_{r,s} \mathcal{S}_\pm^{(r,s)_1} e^{\mp T_{r,s}^{(b)}} + \cdots \tag{4.3}$$

This has the typical form expected from resurgence. The exponential term $e^{\mp T_{r,s}^{(b)}}$ means that this is a non-perturbative contribution in $e^{-S_0}$. Corrections to this formula are indicated by the ellipses and are suppressed by $e^{-S_0}$. They are computable by the non-perturbative topological recursion developed in [63].

The tension $T_{r,s}^{(b)}$ is computed as [11]

$$T_{r,s}^{(b)} = e^{S_0} \int_{z_{(r,s)}^-}^{z_{(r,s)}^+} \omega_{0,1} = \frac{8e^{S_0}(-1)^{r+s}\sin(\pi r b^2)\sin(\pi s b^{-2})}{b^{-2} - b^2}, \tag{4.4}$$

where $\omega_{0,1}(z) = -y(z)dx(z)$. This integral can be thought of as an integral over the corresponding B-cycle in the spectral curve. The overall sign of the tension (4.4) is ambiguous because we could have reversed the orientation of the integral. Thus we have in fact two possible non-perturbative corrections with opposite tensions. We indicated this by the subscript $+$ and $-$ in (4.3). These two contributions correspond to the instanton and the ghost-instanton contributions in the bulk. We chose conventions such that the sign matches the choice on the worldsheet, see (3.6). Notice also that since $b^2 \in i\mathbb{R}$, the tension is *purely imaginary*. Thus the exponentiated tension is a rapidly oscillating function order 1 in $S_0$, and hence depends very sensitively on the parameters.

$\mathcal{S}_\pm^{(r,s)_1}$ is a Stokes constant indicating the choice of Lefschetz thimble of the non-perturbative correction. As a reminder of the notation, the superscript $(r,s)_1$ denotes the type of nodal singularity, with its multiplicity indicated in the subscript, which map to the type and multiplicity of the ZZ-instanton on the string theory side. There is a physically relevant choice that we will discuss below, but from the point of view of resurgence, $\mathcal{S}_\pm^{(r,s)_1}$ are free constants.

The prefactor $\frac{1}{2\pi}\mathcal{B}_{r,s}$ is a one-loop correction mapping to the cylinder diagram on the string theory side. It takes the following form in the matrix model [21],

$$
\begin{aligned}
\mathcal{B}_{r,s} &= \frac{\sqrt{2\pi}e^{\frac{1}{2}S_0}}{z_{(r,s)}^- - z_{(r,s)}^+} \left( \frac{dx}{dz}(z_{(r,s)}^-)\frac{dy}{dz}(z_{(r,s)}^+) - \frac{dx}{dz}(z_{(r,s)}^+)\frac{dy}{dz}(z_{(r,s)}^-) \right)^{-\frac{1}{2}} \\
&= i\,(2\pi T_{r,s}^{(b)})^{-\frac{1}{2}} \left( \frac{\frac{b^{-2}}{r^2} - \frac{b^2}{s^2}}{b^{-2} - b^2} \right)^{\frac{1}{2}}.
\end{aligned}
\tag{4.5}
$$

We chose a convenient branch of the square root. The ambiguity of choosing the branch can be absorbed into the Stokes constant. With this convention, we see that (4.3) matches precisely with the worldsheet result (3.12) and (3.14) with $n = 0$.

**Correlators.**   We can also obtain non-perturbative corrections to correlators as follows. For no operator insertions, $A_{g,0}^{(b)} = \omega_{g,0}^{(b)}$. Resolvents with more insertions can be obtained by applying the loop insertion operator $\mathcal{L}_z$, which acts as a derivation, since it can be written as a formal differential on the matrix model side with respect to the Taylor coefficients of the potential [64]. We have by definition

$$\mathcal{L}_z \omega_{g,n}(z_1,\ldots,z_n) = \omega_{g,n+1}(z_1,\ldots,z_n,z). \tag{4.6}$$

Crucially, this continues to hold in the non-perturbative sector. Thus the leading non-perturbative correction to $\omega_1^{(b)}(S_0; z)$ due to a single nodal singularity of type $(r, s)$ is

$$
\begin{aligned}
\omega_1^{(b)}(S_0; z)_\pm^{(r,s)_1} &= \frac{1}{2\pi} \mathcal{B}_{r,s} \mathcal{S}_\pm^{(r,s)_1} \mathscr{L}_z \cdot e^{\mp T_{r,s}^{(b)}} \\
&= \mp \frac{1}{2\pi} \mathcal{B}_{r,s} \mathcal{S}_\pm^{(r,s)_1} e^{\mp T_{r,s}^{(b)}} \mathscr{L}_z \int_{z_{(r,s)}^-}^{z_{(r,s)}^+} \omega_{0,1}^{(b)} + \cdots \\
&= \mp \frac{1}{2\pi} \mathcal{B}_{r,s} \mathcal{S}_\pm^{(r,s)_1} e^{\mp T_{r,s}^{(b)}} \int_{z_{(r,s)}^-}^{z_{(r,s)}^+} \omega_{0,2}^{(b)}(\bullet, z) + \cdots \\
&= \mp \frac{1}{2\pi} \mathcal{B}_{r,s} \mathcal{S}_\pm^{(r,s)_1} e^{\mp T_{r,s}^{(b)}} \left( \frac{1}{z - z_{(r,s)}^+} - \frac{1}{z - z_{(r,s)}^-} \right) dz + \cdots
\end{aligned}
\tag{4.7}
$$

For a much more complete discussion on the non-perturbative corrections to the resolvents, see [63]. A useful mnemonic for the action of $n$ loop insertion operators acting on a more general multi-instanton sector of the free energy of the matrix model is shown in figure 6. In (4.7), the $+ \cdots$ denote terms that are further suppressed by powers of $e^{-S_0}$. We now transform (4.7) back to the string amplitudes $A_1^{(b)}(S_0; p)_\pm^{(r,s)_1}$. The transformation is an inverse Laplace transformation in the coordinate $w = \sqrt{z}$,

$$
\begin{aligned}
A_1^{(b)}(S_0; p)_\pm^{(r,s)_1} &= \int_\gamma \frac{e^{2\pi i p w}}{4\pi i p} \omega_1^{(b)}(S_0; w^2)_\pm^{(r,s)_1} \\
&= \sum_{\pm, \pm} \operatorname*{Res}_{w = \pm r b \pm s b^{-1}} \frac{e^{2\pi i p w}}{2p} \omega_1^{(b)}(S_0; w^2)_\pm^{(r,s)_1} \\
&= \pm \mathcal{B}_{r,s} \mathcal{S}_\pm^{(r,s)_1} e^{\mp T_{r,s}^{(b)}} \frac{\sin(2\pi r b p) \sin(2\pi s b^{-1} p)}{\pi p}.
\end{aligned}
\tag{4.8}
$$

In particular the ratio is

$$
\frac{A_1^{(b)}(S_0; p)_\pm^{(r,s)_1}}{A_0^{(b)}(S_0)_\pm^{(r,s)_1}} = \pm \frac{2 \sin(2\pi r b p) \sin(2\pi s b^{-1} p)}{p},
\tag{4.9}
$$

which is identified with the disk one-point diagram in the bulk theory, see (3.4).

$n$-**point amplitude.** We may generalize this computation to the $n$-point amplitude by considering the action of $n$ loop insertion operators on the matrix model partition function (see figure 6, with the index $\alpha$ running over a single ZZ-instanton of type $(r, s)$). The leading order contribution comes from the term in which each loop insertion operator acts on the integrated matrix model differential $\omega_{0,1}$, represented by the empty disk diagram in the mnemonic of figure 6, exactly as in (4.7). Therefore, the action of $n$ loop insertion operators precisely reproduces the leading order worldsheet expressions (3.12) and (3.14).

**Multi-instantons.** The non-perturbative contribution from a multi-instanton configuration to the matrix model partition function was also analyzed in [21]. Again, let $\mathcal{I} = \{(r, s)_{\ell_{r,s}}\}$ denote a set of pairs of integers indexing the type of nodal singularity/ZZ-instanton present in our multi-instanton configuration, with their multiplicities indicated by the subscript $\ell_{r,s} \in \mathbb{Z}_{\geqslant 1}$. Specializing to the complex Liouville string, it was shown that integrating a number $\ell_{r,s}$ of

$$\left(e^{-S_0}\mathscr{L}_{z_n}\right)\cdots\left(e^{-S_0}\mathscr{L}_{z_1}\right)\cdot\exp\left(\mp e^{S_0}\sum_\alpha \overset{\alpha}{\bigcirc} + \sum_\alpha\sum_\gamma \overset{\alpha}{\textcircled{\gamma}} + \cdots\right)$$

Figure 6: Mnemonic for the action of $n$ loop insertion operators acting on the $(\sum_\alpha)$-instanton (or -ghost-instanton for the lower sign) sector of the partition function of the matrix model. Here, the index $\alpha$ runs over all the instantons (either of the same or distinct type) in the given instanton sector in consideration. The loop insertion operator acts as a derivation on a given matrix model observable, for instance on the differentials $\omega_{g,n}$. Note that this calculation is entirely from the matrix model side of the duality, but we use worldsheet diagrams as a mnemonic of how the loop insertion operators act on a multi-instanton configuration. A more precise formula may be found in [63]. The leading order contribution, of order 1 in the genus expansion parameter $e^{-S_0}$, comes from the term in which each loop insertion operator acts on the sum over empty disk diagrams. Other contributions give rise to sub-leading contributions in $e^{-S_0}$ in the same instanton sector.

eigenvalues along the Lefschetz thimble of the saddle point $(\mathsf{x}(z_{r,s}^\pm), \mathsf{y}(z_{r,s}^\pm))$, for each $(r,s) \in \mathcal{I}$, results in [21]

$$A_0^{(b)}(S_0)_\pm^{\mathcal{I}} = \mathcal{S}_\pm^{\mathcal{I}}\left(\prod_{(r,s)\in\mathcal{I}} e^{\mp\ell_{r,s}T_{r,s}^{(b)}}(\mathcal{B}_{r,s})^{(\ell_{r,s})^2}\frac{G_2(\ell_{r,s}+1)}{(2\pi)^{\frac{1}{2}\ell_{r,s}(\ell_{r,s}+1)}}\right)\left(\prod_{\substack{(r,s),(r',s')\in\mathcal{I}\\(r,s)<(r',s')}}(\mathcal{C}_{r,s;r',s'})^{\ell_{r,s}\ell_{r',s'}}\right), \quad (4.10)$$

where $\mathcal{B}_{r,s}$ is the same quantity defined in (4.5) that maps to the exponentiated cylinder diagram with identical boundary conditions. The lower signs correspond to the analogous configuration of multi-ghost-instantons. The matrix model quantity $\mathcal{C}_{r,s;r',s'}$, which maps to the exponentiated cylinder diagram with distinct ZZ-instanton boundary conditions on the string theory side of the duality, is given by [21]

$$\begin{aligned}
\mathcal{C}_{r,s;r',s'} &= \frac{\left(z_{(r,s)}^+ - z_{(r',s')}^+\right)\left(z_{(r,s)}^- - z_{(r',s')}^-\right)}{\left(z_{(r,s)}^+ - z_{(r',s')}^-\right)\left(z_{(r,s)}^- - z_{(r',s')}^+\right)} \\
&= \frac{\left((r-r')^2 b^2 - (s-s')^2 b^{-2}\right)\left((r+r')^2 b^2 - (s+s')^2 b^{-2}\right)}{\left((r-r')^2 b^2 - (s+s')^2 b^{-2}\right)\left((r+r')^2 b^2 - (s-s')^2 b^{-2}\right)}.
\end{aligned} \quad (4.11)$$

Therefore, we see that (4.10), together with (4.5) and (4.11), precisely matches the string theory result (3.21) with (3.23) for the case of no external on-shell vertex operator insertions.

In order to include on-shell vertex operator insertions into the multi-instanton contribution, we can again make use of loop insertion operators. The leading-order non-perturbative contribution from the configuration of ZZ-instantons indexed by the set $\mathcal{I}$ to the differential/resolvent $\omega_n^{(b)}$ comes from each loop insertion operator acting on the sum of exponentiated disk diagrams (see figure 6). Denoting this contribution by $\omega_n^{(b)}(S_0;\boldsymbol{z})_\pm^{\mathcal{I}}$, analogously to

that of the string amplitude $\mathsf{A}_n^{(b)}(S_0; \boldsymbol{p})$, we thus obtain that

$$
\begin{aligned}
\omega_n^{(b)}(S_0; \boldsymbol{z})_\pm^{\mathcal{I}} = \mathcal{S}_\pm^{\mathcal{I}} &\left( \prod_{(r,s)\in\mathcal{I}} e^{\mp \ell_{r,s} T_{r,s}^{(b)}} (\mathcal{B}_{r,s})^{(\ell_{r,s})^2} \frac{G_2(\ell_{r,s}+1)}{(2\pi)^{\frac{1}{2}\ell_{r,s}(\ell_{r,s}+1)}} \right) \\
&\times \left( \prod_{\substack{(r,s),(r',s')\in\mathcal{I} \\ (r,s)<(r',s')}} (\mathcal{C}_{r,s;r',s'})^{\ell_{r,s}\ell_{r',s'}} \right) \\
&\times \left( \prod_{j=1}^n \mp \sum_{(r,s)\in\mathcal{I}} \ell_{r,s} \int_{z_{(r,s)}^-}^{z_{(r,s)}^+} \omega_{0,2}^{(b)}(\bullet, z_j) \right) + \cdots
\end{aligned}
\tag{4.12}
$$

Performing the inverse Laplace transformation to recover the actual string amplitude correction $\mathsf{A}_n^{(b)}(S_0; \boldsymbol{p})_\pm^{\mathcal{I}}$ in exactly the same way as in (4.8), we finally obtain that

$$
\begin{aligned}
\mathsf{A}_n^{(b)}(S_0; \boldsymbol{p})_\pm^{\mathcal{I}} = \mathcal{S}_\pm^{\mathcal{I}} &\left( \prod_{(r,s)\in\mathcal{I}} e^{\mp \ell_{r,s} T_{r,s}^{(b)}} (\mathcal{B}_{r,s})^{(\ell_{r,s})^2} \frac{G_2(\ell_{r,s}+1)}{(2\pi)^{\frac{1}{2}\ell_{r,s}(\ell_{r,s}+1)}} \right) \\
&\times \left( \prod_{\substack{(r,s),(r',s')\in\mathcal{I} \\ (r,s)<(r',s')}} (\mathcal{C}_{r,s;r',s'})^{\ell_{r,s}\ell_{r',s'}} \right) \\
&\times \left( \prod_{j=1}^n \sum_{(r,s)\in\mathcal{I}} \ell_{r,s} \frac{\pm 2\sin(2\pi r b p)\sin(2\pi s b^{-1} p)}{p} \right) + \cdots,
\end{aligned}
\tag{4.13}
$$

which precisely matches the string theory result (3.21) with (3.4).

## 4.2 Large genus asymptotics

From the leading non-perturbative corrections to the string amplitudes, we can extract the large genus asymptotics via resurgence. The computation is very similar to what has been explained in [6, 13] and we will be somewhat brief in describing it. Consider first the Borel transform

$$
\widetilde{\mathsf{A}}_n^{(b)}(x; \boldsymbol{p}) = \sum_{g=0}^\infty \frac{x^{2g}}{(2g)!} \mathsf{A}_{g,n}^{(b)}(\boldsymbol{p}).
\tag{4.14}
$$

It is related to the resummed amplitude via Laplace transformation

$$
\mathsf{A}_n^{(b)}(S_0; \boldsymbol{p}) = e^{-(n-3)S_0} \int_0^\infty dx\, e^{-e^{S_0} x} \widetilde{\mathsf{A}}_n^{(b)}(x; \boldsymbol{p}).
\tag{4.15}
$$

The Borel transform has a number of singularities on the Borel plane at $x^* = \pm \hat{T}_{r,s}^{(b)}$ with $T_{r,s}^{(b)} = e^{S_0} \hat{T}_{r,s}^{(b)}$ whose local behaviour determines the leading large $g$ behaviour of $\mathsf{A}_{g,n}^{(b)}(\boldsymbol{p})$. Since the tensions are imaginary, the Borel singularities are all on the imaginary axis. We claim that the local contribution from the instanton and ghost instanton $(r,s)$ takes the form

$$
\widetilde{\mathsf{A}}_n^{(b)}(x; \boldsymbol{p}) \sim i\, \mathcal{S}^{(r,s)_1} \frac{(2\hat{T}_{r,s}^{(b)})^{n-3}(x^2 - (\hat{T}_{r,s}^{(b)})^2)^{\frac{5}{2}-n}}{2\pi^{\frac{3}{2}}\Gamma(\frac{7}{2}-n)} \prod_{i=1}^n \frac{2\sin(2\pi r b p_j)\sin(2\pi s b^{-1} p_j)}{p_j}.
\tag{4.16}
$$

Here we already assumed that $\mathcal{S}^{(r,s)_1} \equiv \mathcal{S}_+^{(r,s)_1} = \mathcal{S}_-^{(r,s)_1}$. This is necessary to obtain an even function, which is obviously required in view of the definition (4.14). We hence get a second argument for the equality of the Stokes constant of the instanton and the ghost instanton.

To confirm (4.16), we deform the contour in (4.15) to go through the Borel plane singularity and evaluate the integral via saddle point approximation. Performing the saddle point evaluation precisely gives back the leading instanton correction.

It is now straightforward to extract the Taylor expansion coefficients from (4.16) and expand for large genus to get

$$
\mathsf{A}_{g,n}^{(b)}(\boldsymbol{p}) \sim i\, \mathcal{S}^{(r,s)_1} \frac{\left(\hat{T}_{r,s}^{(b)}\right)^{\frac{1}{2}}}{\left(-\hat{T}_{r,s}^{(b)}\right)^{\frac{1}{2}}} \frac{\Gamma\left(2g+n-\frac{5}{2}\right)}{2^{\frac{1}{2}}\pi^{\frac{5}{2}}} \left(\frac{\frac{b^2}{r^2}-\frac{1}{b^2 s^2}}{b^2-\frac{1}{b^2}}\right)^{\frac{1}{2}} \left(\hat{T}_{r,s}^{(b)}\right)^{2-2g-n}
$$
$$
\times \prod_{i=1}^{n} \frac{2\sin(2\pi r b p_i)\sin(2\pi s b^{-1} p_i)}{p_i}. \tag{4.17}
$$

The ratio of the square root of the tensions evaluates of course to $i$ or $-i$ and the ambiguity of the branch can be absorbed into the definition of the Stokes constant.

**Effective string coupling.** This large genus asymptotics shows the $(2g)!$ growth that is characteristic for string amplitudes [25]. Moreover, we see that the effective string coupling is

$$
g_s^{\text{eff}} = \frac{1}{\hat{T}_{r,s}^{(b)}}\, e^{-S_0} = \frac{1}{T_{r,s}^{(b)}}. \tag{4.18}
$$

Rather strikingly, this effective string coupling is *imaginary*, so that the terms in the genus expansion are alternating in sign. The origin of this alternating sign is actually simple to track down – we explicitly put it there in the worldsheet description. Indeed, recall from [1] that

$$
C_b^{S^2} = 32\pi^4 \left(\frac{\sin(\pi b^2)\sin(\pi b^{-2})}{b^2-b^{-2}}\right)^2 = \frac{\pi^4}{2}(\hat{T}_{1,1}^{(b)})^2. \tag{4.19}
$$

Thus we essentially multiplied the string coupling by hand by $(\hat{T}_{1,1}^{(b)})^{-1}$. This might seem artificial, but is natural in light of the effective string couplings in the other instanton sectors (4.18). Moreover, we know that we cannot change the phase of the string coupling in the matrix model as this wouldn't lead to a real density of states in the matrix model.

**The leading large order behaviour.** Consider now (4.17) for $r=s=1$. Since $|\hat{T}_{1,1}^{(b)}| < |\hat{T}_{r,s}^{(b)}|$ for $(r,s) \neq (1,1)$, this gives the leading large order behaviour of the string amplitude. (4.17) simplifies to

$$
\mathsf{A}_{g,n}^{(b)}(\boldsymbol{p}) \sim -\mathcal{S}^{(1,1)_1} \frac{\Gamma(2g+n-\frac{5}{2})}{2^{\frac{1}{2}}\pi^{\frac{5}{2}}} (\hat{T}_{1,1}^{(b)})^{2-2g-n} \prod_{i=1}^{n} \frac{2\sin(2\pi b p_i)\sin(2\pi b^{-1} p_i)}{p_i}. \tag{4.20}
$$

Of course, this still leaves the choice of $\mathcal{S}^{(1,1)_1}$ arbitrary. However, we implicitly fixed it above by using that only one side of the saddle point gives a contribution and thus $\mathcal{S}^{(1,1)_1} = \pm\frac{1}{2}$. The sign is easy to fix since we know from the worldsheet that the moduli space integrand is positive and the sign only comes from the leg factors and the string coupling $g_s^{\text{eff}}$. This leads to $\mathcal{S}^{(1,1)_1} = -\frac{1}{2}$. We will also verify this again below.

**Comparison with perturbative amplitudes.** We can check (4.20) directly by checking with the explicitly computed $\mathsf{A}_{g,n}^{(b)}$. In general, this is quite complicated. However, the perturbative amplitudes $\mathsf{A}_{g,n}^{(b)}$ simplify greatly in a large $g$ limit for large (imaginary) values of $b^2$, where

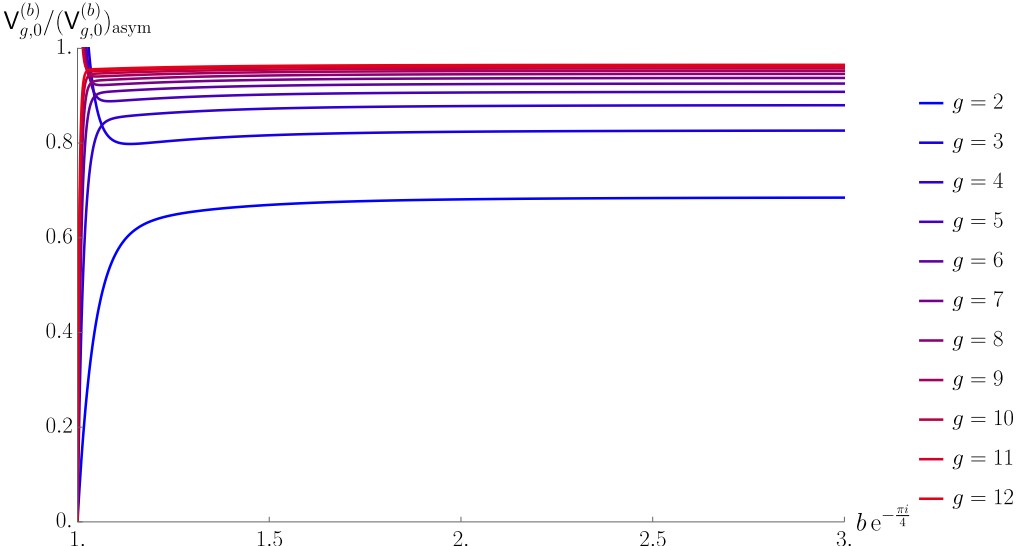

Figure 7: The large $g$ asymptotics of the quantum volumes with imaginary $b^2$. The asymptotics breaks down near $b^2 = i$, but this is not a problem since we are interested in large $b^2$.

the sums over $m_i$ is dominated by the $m_i = 1$ term. Recall that they were expressed as a sum over stable graphs,

$$A_{g,n}^{(b)}(\boldsymbol{p}) = \sum_{\Gamma \in \mathcal{G}_{g,n}^{\infty}} \frac{1}{|\mathrm{Aut}(\Gamma)|} A_{g,n,\Gamma}^{(b)}(\boldsymbol{p}). \tag{4.21}$$

Every contribution $A_{g,n,\Gamma}^{(b)}(\boldsymbol{p})$ is composed as a product over vertices $(g_v, n_v)$, which contain the Virasoro minimal string quantum volumes [13], which behave like $\Gamma(2g_v + n_v - \frac{5}{2})$ in a large $g$ limit. By counting the number of stable graphs, it is straightforward to see that the sum (4.21) is dominated by the trivial graph with a single vertex. We called this limit the semiclassical limit in [2] and already observed the reduction to the $m = 1$ contribution there. We can then use the explicit equation of [2] for the large $g$ asymptotics of the quantum volumes. That equation was valid for $0 < b < 1$, while we are interested in large and imaginary choices of $b^2$. We exploit the $b \to b^{-1}$ invariance of the quantum volumes and replace $b$ by $b^{-1}$ in the formula given in [13] for the large $g$ limit of the quantum volumes. Using the Mathematica code provided in [13], we checked also explicitly that the formula remains valid for imaginary $b^2$, see figure 7. Inserting this asymptotics into the trivial graph with $m = 1$ gives then

$$A_{g,n}^{(b)}(\boldsymbol{p}) \sim \left(-\frac{b}{\sqrt{2}\sin(\pi b^2)}\right)^{2g-2+n} \prod_{j=1}^{n} \sqrt{2}\sin(2\pi b p_j) V_{g,n}^{(b)}(i\boldsymbol{p})$$

$$\sim \frac{\Gamma\left(2g + n - \frac{5}{2}\right)}{2^{\frac{3}{2}}\pi^{\frac{5}{2}}(1 - b^{-4})^{\frac{1}{2}}} \left(\hat{T}_{1,1}^{(b)}\right)^{2-2g-n} \prod_{j=1}^{n} \frac{2\sin(2\pi b p_j)\sin(2\pi b^{-1} p_j)}{p_j}. \tag{4.22}$$

This matches with what we found above, except for the factor $(1 - b^{-4})^{-\frac{1}{2}}$, which is immaterial for large $b^2$. In particular, we confirm the choice of the Stokes constant $\mathcal{S}^{(1,1)_1}$.

## 4.3 Negativities in the density of states

Let us recall from [2] that the density of states in the eigenvalues of the first matrix is given by

$$\rho_0^{(1)}(E) = \frac{2}{\pi}\sinh(-\pi i b^2)\sin\left(-ib^2\mathrm{arccosh}\left(\frac{E}{2}\right)\right). \tag{4.23}$$

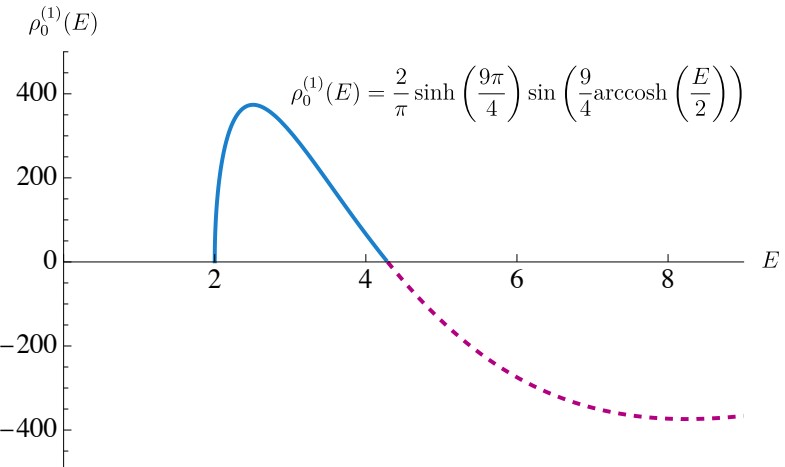

Figure 8: Eigenvalue density for $b = \frac{3}{2}e^{\frac{\pi i}{4}}$. Close to $E \approx 2$ we find the characteristic square root behaviour while the first zero occurs at $E_0 = 2\cos(\pi b^{-2})$. At this stage we need to deal with non-perturbative effects, which in particular necessitate the deformation of the integration contour from the real $E$-axis into the complex $E$ plane.

We also remember that $-ib^2 \in \mathbb{R}_{>0}$ in our parametrization. The lower edge of the energy spectrum is located at $E = 2$ with characteristic square root behaviour near the threshold,

$$\rho_0^{(1)}(E) \approx -\frac{2}{\pi} b^2 \sin(\pi b^2) \sqrt{E-2}. \qquad (4.24)$$

More interestingly, while $\rho_0^{(1)}(E)$ is initially positive close to the edge, it reaches a maximum at $E_{\max} = 2\cos(\frac{1}{2}\pi b^{-2})$. For higher energies, it decreases again and has a simple zero at $E_0 = 2\cos(\pi b^{-2})$, after which the density of states becomes negative. See figure 8 for a representative plot.

There is similarly a density of states for the second matrix, which is derived analogously and takes the form

$$\rho_0^{(2)}(E^{(2)}) = \frac{2}{\pi} \sinh(\pi i b^{-2}) \sin\left( i b^{-2} \mathrm{arccosh}\left( -\frac{E^{(2)}}{2} \right) \right), \qquad (4.25)$$

supported on energies $E^{(2)} \leqslant -2$. It similarly is only initially positive in the region $E_0^{(2)} = -2\cos(\pi b^2) < E^{(2)} < -2$.

This is clearly unphysical. However, we should notice that this negative region in the density of states only affects non-perturbative quantities since it is far away from the branch point that controls the topological recursion.

The physical interpretation of the theory can be salvaged in the same way as the non-perturbative instabilities in JT-gravity [6]. In JT-gravity, the integral over the eigenvalues of the hermitian matrix is deformed away from the real axis in the classically forbidden regime of the eigenvalues. We will discuss below the analogue for the two-matrix model, but let us first review some generalities about the effective potential and its saddle points.

**Effective potential.** Consider the effective potential that is felt by a single pair of eigenvalues $(\lambda, \mu)$ as a consequence of the presence of the other eigenvalues, see [2] for a general discussion about this

$$V_{\mathrm{eff}}(\lambda_i, \mu_i) = V_1(\lambda_i) + V_2(\mu_i) - \lambda_i \mu_i - \frac{1}{N} \sum_{j, j \neq i} \log(\lambda_i - \lambda_j) - \frac{1}{N} \sum_{j, j \neq i} \log(\mu_i - \mu_j). \qquad (4.26)$$

The first derivative at large $N$ can be computed in terms of the functions

$$Y(x) = V_1'(x) - P \int \frac{\rho_0^{(1)}(x')\,dx'}{x - x'}\,, \tag{4.27}$$

$$X(y) = V_2'(y) - P \int \frac{\rho_0^{(2)}(y')\,dy'}{y - y'}\,. \tag{4.28}$$

It reads

$$\partial_x V_{\text{eff}}(x, y) = Y(x) - y\,, \qquad \partial_y V_{\text{eff}}(x, y) = X(y) - x\,. \tag{4.29}$$

We can integrate these equations to compute the effective potential in the large $N$ limit. It is useful to parametrize it through the uniformizing coordinate $z$ on the spectral curve and set $x = \mathsf{x}(z)$, $y = \mathsf{y}(z')$, this gives

$$V_{\text{eff}}(\mathsf{x}(z), \mathsf{y}(z')) = \int d\mathsf{x}(z)\,\mathsf{y}(z) + \int d\mathsf{y}(z')\,\mathsf{x}(z') - \mathsf{x}(z)\mathsf{y}(z') \tag{4.30}$$

$$= 4\cos(\pi b^{-1}\sqrt{z})\cos(\pi b\sqrt{z'}) - \frac{2}{b}\left(\frac{\cos(\pi\hat{Q}\sqrt{z})}{\hat{Q}} + \frac{\cos(\pi Q\sqrt{z})}{Q}\right)$$

$$+ 2b\left(\frac{\cos(\pi\hat{Q}\sqrt{z'})}{\hat{Q}} - \frac{\cos(\pi Q\sqrt{z'})}{Q}\right). \tag{4.31}$$

**Saddle points.** The saddle point equation for the effective potential reads

$$Y(x^*) = y^*\,, \qquad X(y^*) = x^*\,. \tag{4.32}$$

Let us explain that such saddle points correspond to nodal singularities of the spectral curve which in turn are associated to ZZ-instantons. Recall from the definition of the spectral curve (which we reviewed in [2]) that

$$P_0(x, Y(x)) = P_0(X(y), y) = 0\,, \tag{4.33}$$

for a polynomial $P_0$ (before double scaling). Thus $(x^*, y^*)$ is a point on the spectral curve. One may think that the first equation in (4.32) implies the second one. Assuming only the first one, we have $(x^*, Y(x^*)) = (x^*, y^*)$ on the spectral curve. Thus we can parametrize $x^* = \mathsf{x}(z_*^+)$ and $y^* = \mathsf{y}(z_*^+)$. But also $(X(y^*), y^*) = (X(\mathsf{y}(z_*^+)), \mathsf{y}(z_*^+))$ lies on the spectral curve by definition and so

$$(X(\mathsf{y}(z_*^+)), \mathsf{y}(z_*^+)) = (\mathsf{x}(z_*^-), \mathsf{y}(z_*^-))\,. \tag{4.34}$$

Thus we have $\mathsf{y}(z_*^+) = \mathsf{y}(z_*^-)$, but this does not imply $z_*^+ = z_*^-$ since $\mathsf{y}$ is multi-valued. Indeed, they generically lie on different sheets. We get an instanton saddle if two sheets collide as then $z_*^+$ and $z_*^-$ correspond to the same point on the spectral curve.

In the case at hand, saddle points come in a two-parameter family $z_{(r,s)}^{\pm} = (rb \pm sb^{-1})^2$ with $r, s \in \mathbb{Z}_{\geqslant 0}$ as appeared already in (4.2). We note that the change of sign in the density of states precisely coincides with the location of the first ZZ-instanton $z_{(1,1)}^{\pm}$ on the spectral curve. Indeed,

$$\mathsf{x}\left(z_{(1,1)}^{\pm}\right) = 2\cos(\pi b^{-2}) = E_0^{(1)}\,, \qquad \mathsf{y}\left(z_{(1,1)}^{\pm}\right) = -2\cos(\pi b^2) = E_0^{(2)}\,. \tag{4.35}$$

Similarly, higher zeros of the density of states correspond to the general $(r, s)$ instantons.

We now discuss the expansion of the effective potential around the saddle-point. The effective potential evaluated on the saddle points gives the instanton and ghost-instanton tension (3.6)

$$V_{\text{eff}}\left(\mathsf{x}\left(z_{(r,s)}^-\right), \mathsf{y}\left(z_{(r,s)}^+\right)\right) = \hat{T}_{r,s}^{(b)}\,, \qquad V_{\text{eff}}\left(\mathsf{x}\left(z_{(r,s)}^+\right), \mathsf{y}\left(z_{(r,s)}^-\right)\right) = -\hat{T}_{r,s}^{(b)}\,. \tag{4.36}$$

Thus the former possibility corresponds to the instanton saddle, while the latter is identified with the ghost instanton.

We next want to compute the Hessian of the effective potential which arises from expanding around the saddle. It cannot be directly evaluated from the effective potential because one also has to keep corrections from shifting $N$ when pulling out the instantons and from the connected correlators of the Vandemonde determinants. These ingredients together give the general answer for $\mathcal{B}$ quoted in eq. (4.5). We refer to [21] for details. Nonetheless, computing the Hessian is still useful because these corrections only constitute overall prefactors to the matrix.

The Hessian matrix around the ZZ-instanton location in the $z$-coordinate becomes

$$\text{Hess } V_{\text{eff}}\left(\times\left(z_{r,s}^-\right),y\left(z_{r,s}^+\right)\right) = \frac{\pi^2}{8}(b^{-2}-b^2)\hat{T}_{r,s}^{(b)}\begin{pmatrix} -\frac{1}{(rb-sb^{-1})^2} & -\frac{1}{(rb+sb^{-1})(rb-sb^{-1})} \\ -\frac{1}{(rb+sb^{-1})(rb-sb^{-1})} & -\frac{1}{(rb+sb^{-1})^2} \end{pmatrix}. \quad (4.37)$$

The eigenvectors of this matrix are unaffected by the aforementioned scalar corrections and read

$$v = \begin{pmatrix} -\frac{b^2r^2+b^{-2}s^2 \pm \sqrt{2(b^4r^4+b^{-4}s^4)}}{b^2r^2-b^{-2}s^2} \\ 1 \end{pmatrix}. \quad (4.38)$$

Note that the first entry of the eigenvector is real. In particular, in the special case of $b = e^{\frac{\pi i}{4}}$ and $r = s = 1$, the eigenvectors are simply $v = (\pm 1\ 1)^{\mathsf{T}}$. This means that the steepest descent contour near the saddles is entangled between the two matrices.

**Convergence of the matrix integral.**  We now discuss the convergence of the matrix integral and how this is connected to the negativities in the density of states. For this, we integrate out one eigenvalue pair at a time. We are hence interested in the integral

$$\int_\Gamma d\lambda\, d\mu\, e^{-N\sum_i V_{\text{eff}}(\lambda_i,\mu_i)}, \quad (4.39)$$

where $(\lambda,\mu) = (\lambda_N,\mu_N)$. In the large $N$ limit, we can replace the effective potential with its continuum version (4.31) with the caveats mentioned above. We could also consider additional operator insertions in (4.39), but these do not modify the discussion and we will omit them. We started by considering a hermitian matrix model and thus would naively expect that the integration contour is $\Gamma = \mathbb{R}^2$ for each eigenvalue pair. However, just like in ordinary JT-gravity [6], this leads to a divergent integral. Thus, we will have to change the integration contour to get a well-defined non-perturbative completion. This procedure is not unique.

For $\lambda$ and $\mu$ far outside the physical spectrum (i.e. $\lambda \ll 2$ and $\mu \gg -2$), the effective potential is dominated by the term $-\lambda\mu$. Thus the integral behaves like

$$\int_{\lambda \ll 2} d\lambda \int_{\mu \gg -2} d\mu\, e^{N\lambda\mu}, \quad (4.40)$$

which is rapidly convergent. This shows that it was necessary for convergence to have one set of eigenvalues bounded from below in the physical spectrum and the other set of eigenvalues bounded from above.

For $\lambda$ or $\mu$ in the physical spectrum, i.e. where the respective density of states is positive, we also want to keep the contour unchanged in order not to lose the physical interpretation of the integral. However, the integral over the full $\mathbb{R}^2$ would diverge, for example for the region $\lambda,\mu \gg 0$ or $\lambda,\mu \ll 0$. Thus it is necessary to deform the contour in these dangerous regions. There are many choices of such contours corresponding to different non-perturbative completions of the model.

One of the simplest is to consider the steepest descent contour starting at the first ZZ-instanton, which matches with the change of sign in the density of states. Thus for small $\lambda\mu$, we keep the contour unchanged and equal to the $\mathbb{R}^2$ contour, while the parts for large $\lambda\mu$ are deformed into the steepest descent contour passing through the first ZZ-instanton. Thanks to eq. (4.35), the transition is precisely happening when the density of states turns negative. Thus this deformation achieves two things simultaneously: it makes the matrix integral convergent and it removes the negativities in the density of states. It is difficult to describe this contour more precisely because it in particular cannot factorize as the discussion of the eigenvectors around eq. (4.38) showed.

## 5 Discussion

We will now discuss a few interesting open questions and future directions.

**Minimal string.** The trumpet gluing procedure that we discussed in section 2.1 can actually be applied to all string theories, but becomes particularly simple for 2d string theories. By this term we mean all string theories whose worldsheet effective central charge satisfies $c_{\text{eff}} \leqslant 2$, which we already discussed in [2]. Let us consider for concreteness the $(p,q)$ minimal string, whose quantities we denote by a superscript $(p,q)$. In that case, the physical spectrum is finite and is labelled by the Kac-table $(r,s) \in \text{Kac}_{p,q}$ with $1 \leqslant r \leqslant p-1$ and $1 \leqslant s \leqslant q-1$ with $(r,s) \sim (p-r, q-s)$. Inverting the two-point amplitude $A_{0,2}^{(p,q)}((r,s),(r',s'))$ gives rise to a measure on the space of physical states. For the complex Liouville string, this measure was $-2p\,\mathrm{d}p$. For the minimal string, we can normalize string amplitudes such that the two-point amplitude is orthonormal. Then the gluing of trumpets in the minimal string simply reads

$$Z_{g,n}^{(p,q)}(\Psi_1,\ldots,\Psi_n) = \prod_{j=1}^{n} \sum_{(r_j,s_j)\in\text{Kac}_{p,q}} Z_{\text{trumpet}}^{(p,q)}(\Psi_j,(r_j,s_j)) A_{g,n}^{(p,q)}((r_1,s_1),\ldots,(r_n,s_n)). \tag{5.1}$$

Thus we find the gluing prescription for the minimal string developed in [14] to be somewhat misleading, as amputated amplitudes termed $p$-deformed Weil-Petersson volumes were defined using a continuous gluing measure, see also [65].

Such a gluing of trumpets is of course also possible for a full-fledged superstring, but inverting the two-point function and summing over the full physical spectrum becomes a daunting task in this case, which can at best be partially done using level-truncation.

**Open string insertions.** Returning to the complex Liouville string, one can imagine more general observables than the string partition functions $Z_{g,n}^{(b)}(\Psi)$ by also inserting suitable open string vertex operators on the boundaries. These open string vertex operators can be seen as boundary condition changing vertex operators. Thus we can consider $m$ boundary conditions $\Psi^1,\ldots,\Psi^m$ on a single boundary condition with boundary changing operators $\mathcal{O}^{i,i+1}$ in between them. Such general observables could hence be denoted as

$$Z_{g,n}^{(b)}\left(\text{tr}\left(\mathcal{O}_1^{12}\mathcal{O}_1^{23}\cdots\mathcal{O}_1^{m_1,1}\right),\ldots,\text{tr}\left(\mathcal{O}_n^{12}\mathcal{O}_n^{23}\cdots\mathcal{O}_n^{m_n,1}\right)\right). \tag{5.2}$$

We used a trace notation to emphasize cyclic invariance since the operators are inserted on a circle. We suppress the boundary conditions in this notation. See figure 9.

These more general observables are not naturally contained in the matrix model, although many attempts have been made to incorporate them [14, 34, 66, 67].[21] Some of them were

---

[21] In the $c = 1$ string, however, a suitable infinite energy limit of open strings ending on FZZT branes—called the long string limit—are known to correspond to non-singlet sectors of the dual $c = 1$ matrix quantum mechanics [68–70].

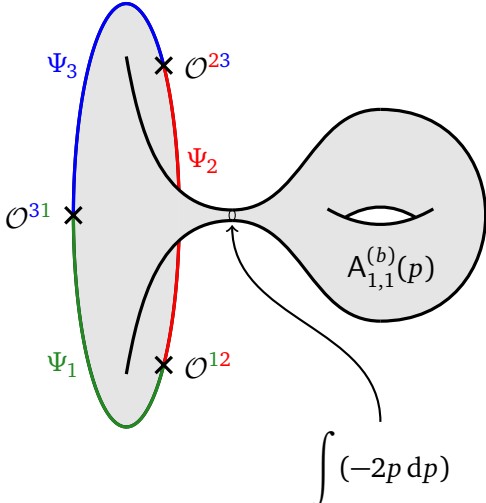

Figure 9: A genus-one contribution to an observable with one asymptotic boundary and three boundary insertions. As in the case without boundary insertions, this is computed by gluing a "trumpet" (punctured disk with the appropriate boundary conditions and boundary insertions) onto the corresponding closed string amplitude. Hence all the nontrivial content of these observables is captured by the trumpet amplitude. This trumpet is characterized by the distinct conformal boundary conditions $\Psi_i$ on the disk together with the data of the corresponding boundary-changing operators $\mathcal{O}^{ij}$. We have used different colors to emphasize the different boundary conditions on the disk.

studied from the worldsheet for the minimal string and related theories in [14]. They tend to be complicated: for example the disk partition function with three boundary insertions with FZZT×ZZ boundary conditions is famously related to the Virasoro fusion kernel [71–74] whose complexity is not reflected in the matrix model.

However, one can treat such configurations with open string insertions just as the partition functions discussed in section 2. In particular, such general observables are determined solely from the knowledge of the partition function on the punctured disk with open string insertions and the perturbative string amplitudes just as in (2.32),

$$
\begin{aligned}
& Z_{g,n}^{(b)}\Big(\mathrm{tr}\big(\mathcal{O}_1^{12}\mathcal{O}_1^{23}\cdots\mathcal{O}_1^{m_1,1}\big),\ldots,\mathrm{tr}\big(\mathcal{O}_n^{12}\mathcal{O}_n^{23}\cdots\mathcal{O}_n^{m_n,1}\big)\Big) \\
& = \int_\gamma \left(\prod_{j=1}^n (-2p_j\,\mathrm{d}p_j)\, Z_{\mathrm{trumpet}}^{(b)}\big(\mathrm{tr}\big(\mathcal{O}_j^{12}\mathcal{O}_j^{23}\ldots\mathcal{O}_j^{m_j,1}\big),p_j\big)\right) A_{g,n}^{(b)}(p_1,\ldots,p_n)\,.
\end{aligned}
\tag{5.3}
$$

This works because even though adding boundary vertex operators creates new moduli, it doesn't change the mapping class group of the surface. In particular, there is no mapping class group element that acts non-trivially on the simple closed curve along which the trumpet is glued and our comments of section 2.1 still hold.

Thus such observables do not contain new gravitational information which motivates why they are not directly captured by the matrix model. The partition function of the punctured disk should be computable by gauge theory techniques, since there is no mapping class group to worry about. While we have not tried this for the complex Liouville string, such an evaluation for the disk without punctures is known in the case of JT-gravity in the form of diagramatic Feynman rules [75–77].

**Subleading non-perturbative corrections.** We have computed the leading non-perturbative corrections from a given instanton sector. However, each such non-perturbative sector has a whole tower of perturbative corrections on top of the leading non-perturbative correction. On the matrix model side, these perturbative corrections are captured by the so-called non-perturbative topological recursion [63]. The computation on the bulk side involves the evaluation of more complicated bulk manifolds with ZZ-boundary conditions. For example in the case of the one-instanton correction, we have

$$
A_n^{(b)}(S_0; \boldsymbol{p})_+^{(r,s)_1} = \exp\left( e^{S_0} \, \bullet + \, \circ \right) \Bigg[ \, \otimes \cdots \otimes
$$

$$
+ e^{-S_0} \Bigg( \otimes \cdot \otimes \cdots \otimes + \otimes \cdot \otimes \cdots \otimes
$$

$$
+ \Big( \otimes + \otimes \Big) \otimes \cdots \otimes + \otimes \cdot \otimes \cdots \otimes
$$

$$
+ \otimes \cdot \otimes \cdots \otimes \Bigg) + \cdots \Bigg]. \tag{5.4}
$$

Such diagrams are naively divergent when applying the gluing formula (2.32) and naive analytic continuation does not lead to correct results. It would be interesting to explore whether such diagrams can also be computed using a modified gluing formula and matched with the non-perturbative topological recursion of [63].

## Acknowledgments

We would like to thank Raghu Mahajan, Marcos Mariño, Maximilian Schwick, Ashoke Sen and Xi Yin for fruitful discussions and comments.

**Funding information** SC, LE and BM thank l'Institut Pascal at Université Paris-Saclay, with the support of the program "Investissements d'avenir" ANR-11-IDEX-0003-01, and SC and VAR thank the Kavli Institute for Theoretical Physics (KITP), which is supported in part by grant NSF PHY-2309135, for hospitality during the course of this work. VAR is supported in part by the Simons Foundation Grant No. 488653, by the Future Faculty in the Physical Sciences Fellowship at Princeton University, and a DeBenedictis Postdoctoral Fellowship and funds from UCSB. SC is supported by the U.S. Department of Energy, Office of Science, Office of High Energy Physics of U.S. Department of Energy under grant Contract Number DE-SC0012567 (High Energy Theory research), DOE Early Career Award DE-SC0021886 and the Packard Foundation Award in Quantum Black Holes and Quantum Computation. BM gratefully acknowledges funding provided by the Sivian Fund at the Institute for Advanced Study and the National Science Foundation with grant number PHY-2207584.

## A  BCFT in Liouville theory

Here we review the standard families of conformal boundary conditions in Liouville CFT. They will play a role in the worldsheet formulation of thermal partition functions and of ZZ-instantons. The simplest two families of conformal boundary conditions in Liouville CFT are

ZZ boundary conditions [15], which may loosely be thought of as Dirichlet-type, and FZZT boundary conditions [78, 79], which may be thought of as Neumann-type. Throughout the following, we adopt the CFT normalization conventions of [13].

**ZZ boundary condition.** The ZZ boundary condition is labelled by a degenerate representation of the Virasoro algebra [15]. By definition, the Hilbert space on the strip with the $(r, s)$ and $(1, 1)$ boundary conditions on each end is spanned by a single scalar Virasoro primary, namely a degenerate primary with conformal weight $h_{r,s}$ (together with its (holomorphic) Virasoro descendants). By considering the modular covariance of the cylinder diagram, one may deduce the main nontrivial piece of OPE data (the disk one-point function of a bulk primary) as follows.

In canonical quantization on the half-infinite cylinder or, equivalently, in radial quantization on the unit disk, the ZZ boundary states can be represented in terms of the Ishibashi states $|V_p\rangle\rangle$ associated with the primaries in the bulk spectrum as

$$|ZZ^{(b)}_{(r,s)}\rangle = -i \int_0^{-i\infty} dp\, \rho_b(p)\psi^{(b)}_{(r,s)}(p)|V_p\rangle\rangle, \qquad \rho_b(p) = 4\sqrt{2}\sin(2\pi b p)\sin(2\pi b^{-1}p), \quad \text{(A.1)}$$

where the function $\psi^{(b)}_{(r,s)}(p)$ is the disk one-point function of the bulk primary $V_p$ in the presence of the $(r, s)$ ZZ boundary condition and $\rho_b(p)$ is the OPE measure in the conventions of [1]. The integration contour corresponds to the spectrum of the theory (2.6). We take the Ishibashi states to be normalized as in (2.8a).

The cylinder diagram with the $(r, s)$ and $(1, 1)$ ZZ boundary conditions assigned to the two ends is computed as follows

$$\langle ZZ^{(b)}_{(r,s)}|\, e^{-\pi t(L_0+\widetilde{L}_0-\frac{c}{12})}\,|ZZ^{(b)}_{(1,1)}\rangle = -i \int_0^{-i\infty} dp\, \rho_b(p)\psi^{(b)}_{(r,s)}(p)\psi^{(b)}_{(1,1)}(p)\chi^{(b)}_p(it)$$

$$\stackrel{!}{=} \mathrm{Tr}_{\mathcal{H}_{(r,s),(1,1)}}\, e^{-\frac{2\pi}{t}(L_0-\frac{c}{24})} = \chi^{(b)}_{(r,s)}\Big(\frac{i}{t}\Big),$$

where

$$\chi^{(b)}_{(r,s)}(\tau) = \frac{q^{-\frac{1}{4}(rb+sb^{-1})^2} - q^{-\frac{1}{4}(rb-sb^{-1})^2}}{\eta(\tau)}, \qquad q = e^{2\pi i\tau}, \quad \text{(A.2)}$$

is the torus character of the $(r, s)$ degenerate representation of the Virasoro algebra. The first line of (A.2), referred to as the closed-string channel, is computed by expanding the conformal boundaries into Ishibashi states using (A.1). The second line of (A.2), referred to as the open-string channel, admits the interpretation as a trace over the Hilbert space of the CFT on the strip with periodic length $\frac{2\pi}{t}$. Hence (A.2) fixes the bulk one-point function of $V_p$ on the disk with $(r, s)$ ZZ boundary condition to be

$$\psi^{(b)}_{(r,s)}(p) = \frac{4\sqrt{2}\sin(2\pi r b p)\sin(2\pi s b^{-1}p)}{\rho_b(p)}. \quad \text{(A.3)}$$

Note that this modular bootstrap of the cylinder diagram does not determine the overall sign of the disk 1-point function (A.3).[22] ZZ boundary states defined with an additional negative sign [80, 81], $-|ZZ^{(b)}_{(r,s)}\rangle$, have been recently understood to give rise to "ghost instanton" contributions [23, 24, 82, 83] to closed string scattering amplitudes. They are relevant in the

---

[22]In a unitary conformal field theory, such as Liouville CFT with $b \in \mathbb{R}$, one can always work in a basis of Virasoro highest weight states such that the different structure functions are real-valued.

complex Liouville string and will be discussed in section 3.3. Note also that (A.3) does not yet contain the contribution from the leg-pole factor which still has to be taken into account when computing one-point functions in the full string theory. Similarly, the more general cylinder diagram with distinct ZZ boundary conditions may be evaluated by expanding the boundary states into Ishibashi states (A.1), resulting in a sum over degenerate Virasoro characters

$$\langle ZZ_{(r,s)}^{(b)}|\, e^{-\pi t(L_0+\widetilde{L}_0-\frac{c}{12})}\,|ZZ_{(r',s')}^{(b)}\rangle = \sum_{r''\overset{2}{=}|r-r'|+1}^{r+r'-1} \sum_{s''\overset{2}{=}|s-s'|+1}^{s+s'-1} \chi_{(r'',s'')}^{(b)}\left(\tfrac{i}{t}\right), \tag{A.4}$$

where the notation $\overset{2}{=}$ indicates that the variable increases in steps of 2. For the $(r,s)$ ZZ boundary condition, there is no further nontrivial BCFT data.

**FZZT boundary conditions.**     The second type of conformal boundary conditions we will need comes in a family known as the FZZT$(u)$ boundary conditions [78,79] labeled by a single continuous parameter $u$, that will be described shortly.[23] In this case, by definition the spectrum on the strip with $(1,1)$ ZZ boundary condition and FZZT boundary condition consists of a single Virasoro primary of non-degenerate weight $h = \frac{Q^2}{4} - u^2$ together with its Virasoro descendants.

Again, consideration of the modular covariance of the cylinder diagram fixes the first piece of BCFT data, namely the disk one-point function $\Psi^{(b)}(u;p)$ of a bulk primary $V_p$ in the presence of an FZZT$(u)$ boundary condition. Expanding the corresponding boundary state in terms of Ishibashi states,

$$|FZZT^{(b)}(u)\rangle = -i\int_0^{-i\infty} dp\, \rho_b(p)\psi^{(b)}(u;p)|V_p\rangle\!\rangle, \tag{A.5}$$

the cylinder diagram with $ZZ_{(1,1)}$ boundary conditions on one end and FZZT$(u)$ boundary conditions on the other is evaluated in the closed- and open-string channels respectively as

$$\langle ZZ_{(1,1)}^{(b)}|\, e^{-\pi t(L_0+\widetilde{L}_0-\frac{c}{12})}|FZZT^{(b)}(u)\rangle = -i\int_0^{-i\infty} dp\, \rho_0^{(b)}(p)\psi_{(1,1)}^{(b)}(p)\psi^{(b)}(u;p)\chi_p^{(b)}(it)$$

$$\overset{!}{=} \mathrm{Tr}_{\mathcal{H}_{(1,1),u}} e^{-\frac{2\pi}{t}(L_0-\frac{c}{24})} = \chi_u^{(b)}\left(\tfrac{i}{t}\right). \tag{A.6}$$

This in turn fixes the bulk one-point function to be

$$\psi^{(b)}(u;p) = -\frac{2\sqrt{2}\cos(4\pi up)}{\rho_b(p)}. \tag{A.7}$$

Note that the FZZT parameter $u$ plays the same role as the parameter $p$, which labels the Liouville momenta and the conformal weights of Liouville vertex operators. Hence, in this paper we will be interested in the locus

$$u \in e^{-\frac{\pi i}{4}}\mathbb{R}_+, \tag{A.8}$$

as is the case for bulk vertex operators [1].

On the other hand, the spectrum on the strip with distinct FZZT$(u_1)$ and FZZT$(u_2)$ boundary conditions on each end consists of a continuous spectrum of boundary Virasoro primaries $\psi_p^{u_1,u_2}$ (and their descendants) labelled by a Liouville momentum $p$ with conformal weights $h_p = \frac{Q^2}{4} - p^2$. Furthermore, for the FZZT boundary condition there are additional nontrivial BCFT data, namely the bulk-to-boundary two-point function and the boundary three-point function on the disk [73,84]. However, we will not make use of these further data in this paper.

---

[23]In the literature it is conventional to label the FZZT boundary conditions by a parameter $s$. Here we are using $u$ to avoid confusion with the integer that is used to characterize the ZZ boundary conditions.

# B   A check of the factorization condition

In this appendix we provide a simple check of the factorization condition (2.30) for the case in which we glue a single boundary to a sphere $n$-point string amplitude $A^{(b)}_{g=0,n}(\boldsymbol{p})$. In this case, the LHS of (2.30) is

$$\frac{32\pi^4 C^{(b)}_{\mathrm{D}^2}}{C^{(b)}_{\mathrm{S}^2} C^{(b)}_{\mathrm{D}^2}} \left( \frac{\sin(\pi b^2)\sin(\pi b^{-2})}{b^2 - b^{-2}} \right)^2 . \tag{B.1}$$

The normalization constant $C^{(b)}_{\mathrm{S}^2}$ associated with the sphere topology was determined in [1] and is given by

$$C^{(b)}_{\mathrm{S}^2} = 32\pi^4 \left( \frac{\sin(\pi b^2)\sin(\pi b^{-2})}{b^2 - b^{-2}} \right)^2 . \tag{B.2}$$

Therefore, (B.1) is precisely equal to 1.

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
