# Peer review of "The complex Liouville string: worldsheet boundaries and non-perturbative effects"

_SciPost Physics, doi:SciPost Phys. 19, 034 (2025)_

## Round 1 · Referee Report · Anonymous (Referee 1) · 2025-5-30

Report

The authors have provided adequate comments on my (and the other referees') questions.

Their new version of the paper has the typos corrected, but as far as I can tell, there is no new clarification on the negativity of the density of states in their actual paper, whereas all three referees pointed out subtleties and confusions on this delicate issue. I believe the authors should still adjust their text a bit and add some further clarification to anticipate some of these confusions that other readers will most likely experience as well. I still think the authors should do this, but I am going to recommend the paper for publication in any case.

Recommendation

Publish (easily meets expectations and criteria for this Journal; among top 50%)

---

## Round 1 · Referee Report · Anonymous (Referee 2) · 2025-6-4

Report

I thank the authors for their response. The authors have made appropriate changes in response to my comments, and the paper is ready for publication.

I recommend that this paper be published in SciPost.

Recommendation

Publish (easily meets expectations and criteria for this Journal; among top 50%)

---

## Round 1 · Referee Report · Anonymous (Referee 3) · 2025-7-3

Report

I thank the authors for addressing the questions and comments in my previous report. I strongly recommend the paper for publication.

Recommendation

Publish (easily meets expectations and criteria for this Journal; among top 50%)

---

## Round 1 · Author Response

Dear editor,

We thank the referees for their careful reading of and helpful comments and suggestions about our draft. Below we respond to the referees' questions (as enumerated in their reports) and describe how we have accordingly modified the draft in turn.

---

## Round 1 · List of Changes

\textbf{Response to referee 1:}

We thank the reviewer for their supportive comment, reading of the manuscript, and for raising the interpretational question regarding the spectral density.

We believe that the results presented in sections 3 and 4 provide strong evidence for the interpretation put forward in our paper. In particular, we perform a first-principles string-theoretic calculation of the leading ZZ-instanton correction for each type of $(r,s)$ instanton and their multi-instanton generalizations. These string calculations precisely match independent computations in the non-perturbative sector of the dual spectral curve.

The agreement between individual saddles (or thimbles) on both sides, together with the fact that the leading ZZ-instanton corresponds to the point at which the leading-order perturbative spectral density becomes negative, and that this same point is where the open string tachyon develops, all provide compelling support for this interpretation. Further tests of this interpretation, such as matching the sub-leading corrections around each $(r,s)$ ZZ-instanton saddle, would be important future research, which is in progress.

We also agree with the reviewer that the choice of non-perturbative integration contour, i.e., the sum over thimbles, is a separate and more subtle issue, which we did not attempt to resolve in this work. This amounts to a choice of Stokes constants, whose determination from first principles is a difficult problem in resurgent analyses. What we have shown is that a deformation of the integration contour away from the real energy axis becomes necessary precisely at the point where the spectral density turns negative.

Relatedly, we should note that we are not certain that there is a ``canonical" non-perturbative completion among the many distinct non-perturbative completions defined by distinct choices of Stokes data.

Regarding the reviewer’s question, we should clarify what we mean by the negativity affecting only ``non-perturbative quantities." True observables of the complex Liouville string are its amplitudes, not directly the spectral density of states. What we mean is that perturbative amplitudes (or resolvents, in the dual spectral curve side), with genus $g$ and $n$ punctures, are not affected by these instanton effects. On the spectral curve side of the duality, these can be reliably computed through topological recursion on the spectral curve. [Additionally, the recent paper 2501.01486 provides a compelling interpretation of the integral of this spectral density truncated to its first zero. This integral provides an effective number of states of the model, which was then nontrivially argued to reproduce the leading de Sitter entropy in 3d.]

Finally, we thank the reviewer for catching the typo in the caption of Figure 8 on page 42. We have corrected the incomplete sentence in the revised version.

\textbf{Response to referee 2:}

We thank the referee for their careful reading and useful comments. Let us answer the points raised in turn and how we addressed them in a revised manuscript:

\begin{enumerate}
\item We don't agree that this calls into question the whole framework of the instanton expansion. There are several situations when non-perturbative effects can be of order 1 due to the instanton effect being purely imaginary, even in more `standard' theories. For example, one can compute the non-perturbative correction to the density of states in JT-gravity. In the allowed region $E>0$, this correction is of order 1 (compared to the leading term which is of order $e^{S_0}$) and rapidly oscillating, exactly as in the instanton effects in this paper, while they are (doubly) exponentially suppressed in the forbidden region $E<0$. Instantons with imaginary tensions are also quite standard in resurgence discussions. Once one considers the complex Borel plane, the existence of instantons with non-positive tensions appears in the form of Stokes lines away from the positive real axis. In those cases, these instantons may or may not contribute to non-perturbative quantities, depending on the choice of Stokes constants.

Let us also mention that in this case, there is a simple way to see that the instanton tension has to be imaginary (that we also mentioned in the paper). From the $b$-dependence of the instanton tension, we see that the sign is flipped under the replacement $b \to b^{-1}$ (that is just a redundancy of the description of the theory). So in order for the non-perturbative quantity to inherit this redundancy, we \emph{have} to include the corresponding ghost instantons with opposite tensions. It would be fare more worrying if the tensions would be real since then instanton effects would be exponentially large, which of course would be a problem.

We agree that the leading density of states going negative is more puzzling and potentially more problematic, even though we gave as far as we can see a consistent resolution in the paper. Let us however mention that non-perturbative corrections to the density of states are still suppressed by a factor $e^{-S_0}=g_s^{-1}$, even if we neglect the rapidly oscillating nature. Such a correction is furthermore precisely expected since it is the hallmark of the theory coming from a matrix model with a discrete spectrum of eigenvalues. This was discussed in a number of papers in the literature in the context of JT-gravity and others. Hence at least for large $S_0$, we don't expect that the non-perturbative effects can make the density of states positive of the whole range. This is already impossible because of their oscillating nature and if we coarse grain slightly over the eigenvalues, they become even smaller.

\item The characters implicitly depend on $b$, since the definition of the Liouville momentum does. If we would write it in terms of the conformal weight, $b$ would appear explicitly. Hence we kept the notation for consistency and made a comment below the equation.

\item We added an explanation underneath equation (2.42). We expand the trigonometric functions into exponentials and then define the contour $\gamma$ as an outgoing ray in the complex plane such that the integral converges. With this convention $\int_\gamma\mathrm{d} p\, p\, \mathrm{e}^{a p}=a^{-2}$ for arbitrary complex $a$.

\item The lower bound is a convention. It relies on our convention of the normalization of the spectral curve that we discussed in 2410.07345. Changing this normalization is essentially equivalent to changing the string coupling, which in turn can be absorbed into $S_0$. We found this convention particularly simple, but it doesn't have any deeper meaning.

\item $b^2$ is in fact purely imaginary and thus in particular never rational which is what the referee seems worried about. A simple alternative way to see this is to notice that the trumpet partition functions for these boundary conditions are all different and thus they cannot be equivalent.

\item A priori, $u$ can be any complex number. The open string spectrum between the FZZT boundary and the ZZ$_{(1,1)}$ boundary consists of a single Virasoro character of Liouville momentum $p=u$. Hence, in standard Liouville theory one usually picks $u \in i \mathbb{R}$ so that the conformal weight of this Virasoro character is real and bounded from below by $\frac{c-1}{24}$. In this work, it would perhaps be natural to rotate the FZZT parameter to be in a 45 degree angle as we did for the vertex operators (so that the Virasoro representation in the open string channel has conformal weight in $\frac{1}{2}+i \mathbb{R}$, i.e.\ the principal series). As far as we can see this however is not actually necessary and any boundary condition is consistent. As a case in point, the resolvent boundary condition that we also introduce is related to a derivative of this FZZT boundary condition. Here, it is natural that the FZZT parameter takes any complex value since it is identified with the coordinate on the spectral curve.

In order to keep the discussion somewhat minimal and not confuse the reader, we hence only added a sentence below (2.57) that $u$ can in principle be any complex number. See the point below about the issue of $u$ and $i u'$.

\item We agree that the $\text{FZZT}(u) \times \text{FZZT}(u')$ boundary condition that we mention was treated in a somewhat cavalier way. In particular, the point about the extra $\rho_b(p)$ in the denominator is a very valid one and prevents one from writing the wavefunction as a sum of $\text{FZZT}(u) \times \text{ZZ}_{(1,1)}$ wavefunctions as we originally claimed. This cannot be possible because the $\rho_b(p)$ in the denominator introduces a triple pole at $p=0$, while the $\text{FZZT}(u) \times \text{ZZ}_{(1,1)}$ wavefunctions only have single poles.

In fact, we investigated this boundary condition further and found that it behaves quite differently than the $\mathrm{FZZT}(u) \times \mathrm{ZZ}_{(1,1)}$ boundary condition. For example, the disk two-point function with this boundary condition is logarithmically divergent from the closed string degeneration because of this triple pole. One can also see this in the gluing formula: the integral over $p$ in eq.~(2.32) diverges near $p=0$ because of the triple pole. For the purposes of the present work, this is not a good boundary condition as it doesn't allow one to define the observables that we discuss in the paper. It leads to non-vanishing tadpoles, which indicates that it backreacts on the background and one should expand around a new corrected background.

In any case, we fortunately don't need this boundary condition anywhere else in the paper. For these reasons, we have decided to remove the paragraph about the $\mathrm{FZZT}(u) \times \mathrm{FZZT}(u')$ boundary condition.

\item Eq.\ (2.61) (in the old manuscript) is based on the usual Laplace inversion formula in which the contour $\Gamma$ is chosen as we described. It seems that the referee is worried about cases where the inversion formula doesn't converge. In the case at hand in eq.~(2.63), as long as $|b p|<\frac{1}{2}$, the integrand decays for large $|\text{Im }x|$ and the integral is independent of the choice of offset in the contour $\Gamma$. For larger values of $b p$, this is formally still true, but requires one to interpret the integral distributionally. We are not sure whether this is the issue that the referee had in mind, but we added the assumption $|b p|<\frac{1}{2}$ to the computation in footnote 12. The result for larger values $b p$ can be obtained by analytic continuation, by interpreting the integral distributionally or by bending the contour $\Gamma$ slightly to the left.
\end{enumerate}

\textbf{Response to referee 3:}

We thank the referee for their careful reading of the paper and for their comments and questions. Here we respond to their questions and describe how we have addressed their comments in the updated draft.

\begin{enumerate}
\item Our perspective on this question is that the $((r,s),(1,1))$ conformal boundary conditions are in an appropriate sense the ``irreducible'' boundary conditions, and hence it is natural that these are the ones captured by the one-instanton sector of the matrix model. The reason for this is that the basic BCFT data for the more general $((k,\ell),(m,n))$ boundary conditions can be expressed in terms of that of $((r,s),(1,1))$ boundary conditions. For example we have
\begin{equation}
\mathsf{Z}_{\rm trumpet}^{(b)}(((k,\ell),(m,n)),p) = \sum_{r\overset{2}{=}|k-m|+1}^{k+m-1}\sum_{s\overset{2}{=}|\ell-n|+1}^{\ell+n-1}\mathsf{Z}_{\rm trumpet}^{(b)}(((r,s),(1,1)),p)\, .
\end{equation}
We have modified the discussion below equation (3.1) to emphasize this point.

\item We thank the referee for the insightful question and for the interesting comments about the situation in the minimal string. We of course agree that the negativities exhibited by the leading density of eigenvalues are puzzling. The honest answer is that we do not have a particularly sharp vision for the details of how the non-perturbative completion --- which involves a determination of the non-perturbative contour of integration in terms of a sum over Lefschetz thimbles --- is realized in practice. We know that for sufficiently large eigenvalues the eigenvalue contour must be deformed in order for the matrix integral to converge, and our analysis indicates that it is natural to deform it along the steepest descent contour associated with the saddle corresponding to the first ZZ-instanton, which occurs precisely when the spectral density turns negative. But it is difficult to describe more explicitly than this, partly because the steepest descent contour does not factorize.

\item We thank the referee for catching these typos. They have been corrected in the new version.
\end{enumerate}

---

## Editorial Decision

published